# Post-meiotic mechanism of facultative parthenogenesis in gonochoristic whiptail lizard species

David V Ho[1,2†], Duncan Tormey[3†‡], Aaron Odell[1], Aracely A Newton[3§],
Robert R Schnittker[3], Diana P Baumann[3], William B Neaves[3],
Morgan R Schroeder[3], Rutendo F Sigauke[3], Anthony J Barley[4], Peter Baumann[1,2,5]*

[1]Department of Biology, Johannes Gutenberg University, Mainz, Germany; [2]Institute of Quantitative and Computational Biosciences, Johannes Gutenberg University, Mainz, Germany; [3]Stowers Institute for Medical Research, Kansas City, United States; [4]School of Mathematical and Natural Sciences, Arizona State University–West Valley Campus, Glendale, United States; [5]Institute of Molecular Biology, Mainz, Germany

*For correspondence:
peter@baumannlab.org

†These authors contributed equally to this work

Present address: ‡Synthego Corporation, Redwood City, United States; §Missouri Western State University, Saint Joseph, United States

Competing interest: The authors declare that no competing interests exist.

**Abstract** Facultative parthenogenesis (FP) has historically been regarded as rare in vertebrates, but in recent years incidences have been reported in a growing list of fish, reptile, and bird species. Despite the increasing interest in the phenomenon, the underlying mechanism and evolutionary implications have remained unclear. A common finding across many incidences of FP is either a high degree of homozygosity at microsatellite loci or low levels of heterozygosity detected in next-generation sequencing data. This has led to the proposal that second polar body fusion following the meiotic divisions restores diploidy and thereby mimics fertilization. Here, we show that FP occurring in the gonochoristic *Aspidoscelis* species *A. marmoratus* and *A. arizonae* results in genome-wide homozygosity, an observation inconsistent with polar body fusion as the underlying mechanism of restoration. Instead, a high-quality reference genome for *A. marmoratus* and analysis of whole-genome sequencing from multiple FP and control animals reveals that a post-meiotic mechanism gives rise to homozygous animals from haploid, unfertilized oocytes. Contrary to the widely held belief that females need to be isolated from males to undergo FP, females housed with conspecific and heterospecific males produced unfertilized eggs that underwent spontaneous development. In addition, offspring arising from both fertilized eggs and parthenogenetic development were observed to arise from a single clutch. Strikingly, our data support a mechanism for facultative parthenogenesis that removes all heterozygosity in a single generation. Complete homozygosity exposes the genetic load and explains the high rate of congenital malformations and embryonic mortality associated with FP in many species. Conversely, for animals that develop normally, FP could potentially exert strong purifying selection as all lethal recessive alleles are purged in a single generation.

## Editor's evaluation

In this valuable paper, convincing evidence is provided for the production of facultatively parthenogenetic whiptail lizards through a gametic duplication. The audience for the work will be broad, given that parthenogenesis is such a fascinating topic.

## Introduction

Incidences of facultative parthenogenesis (FP) have been reported to occur in diverse vertebrate clades including bony fish (*Lampert et al., 2007*), sharks (*Chapman et al., 2007*; *Chapman et al., 2008*; *Feldheim et al., 2010*; *Dudgeon et al., 2017*), snakes (*Dubach et al., 1997*; *Groot et al., 2003*; *Booth and Schuett, 2011*; *Germano and Smith, 2010*; *Booth et al., 2012*; *Kinney et al., 2013*; *Booth et al., 2014*; *Allen et al., 2018*; *Card et al., 2021*), lizards (*Lenk et al., 2005*; *Watts et al., 2006*; *Kratochvíl et al., 2020*), crocodilians (*Booth et al., 2023*), and birds (*Bartelmez and Riddle, 1924*; *Olsen and Marsden, 1954*; *Sarvella, 1973*; *Schut et al., 2008*; *Ramachandran and McDaniel, 2018*; *Parker et al., 2010*). The phenomenon was originally mistaken for long-term sperm storage occurring in zoo environments where females were housed without current or recent access to conspecific males. The most parsimonious explanation was, therefore, that the animal had previously been in contact with a male and that stored sperm was responsible for delayed fertilization (*Booth and Schuett, 2011*; *Holt and Lloyd, 2010*; *Sever and Hamlett, 2002*). However, more recent studies involving microsatellite (MS) and/or amplified fragment length polymorphism (AFLP) analyses revealed no paternal contributions, as all alleles detected in the offspring were only found in the maternal ancestors (*Groot et al., 2003*; *Booth and Schuett, 2011*; *Shibata et al., 2017*; *Ryder et al., 2021*; *Levine et al., 2024*). Females with no access to males producing solely male (ZW systems) or female (XY systems) offspring that only harbor maternal genetic markers are now considered hallmarks of facultative parthenogenesis. Nevertheless, clear examples of long-term sperm storage have also been documented in the recent literature (*Levine et al., 2021*), underscoring the need for molecular methods such as MS analysis or sequencing data to elucidate the underlying mechanisms. Originally thought to only occur in captivity, more recent reports indicate that FP occurs in natural populations as well (*Booth et al., 2012*; *Fields et al., 2015*). Serious concerns have been raised by conservation biologists, as species with dwindling population densities, including the endangered species Komodo dragon (*Watts et al., 2006*), small tooth sawfish (*Fields et al., 2015*), California condor (*Ryder et al., 2021*), and American crocodile (*Booth et al., 2023*) are overrepresented among reports of FP.

While overrepresentation could be a consequence of an increased likelihood of detection in species that are the subject of intense research and conservation efforts, the observations raise the question if FP is an adaptive trait aiding in the colonization of new areas and mitigating the effects of population bottlenecks or is simply a neutral trait (*Fields et al., 2015*). The adaptive trait hypothesis would of course require successful reproduction of FP animals either sexually or parthenogenetically, which to date has only been documented in a few cases (*Kratochvíl et al., 2020*; *Straube et al., 2016*). At the same time, the association of FP with increased homozygosity constitutes a concern for conservation biology, as an increase in FP within dwindling populations further accelerates the loss of genetic diversity, exposes deleterious alleles, and could compromise efforts to maintain the existing gene pool in selective breeding programs (*Groot et al., 2003*; *Booth and Schuett, 2016*). FP is also being studied as a desirable outcome in the commercial production of poultry. However, examination of tens of thousands of unfertilized eggs from several different avian species and strains has not resulted in economically sustainable hatching rates thus far. One of the highest hatch rates for unfertilized eggs is seen in the Beltsville small white turkey with a rate of 0.88% (*Ramachandran and McDaniel, 2018*; *Olsen, 1975*). A better understanding of the triggers and molecular mechanisms underlying FP and the fitness of the resulting offspring are, therefore, needed in a variety of contexts. These include: to understand a fundamental biological mechanism and its significance in vertebrate evolution, to aid in conservation efforts including captive breeding programs, and to possibly harness FP in an agricultural context (*Ryder et al., 2021*).

The high level of homozygosity observed in animals produced by FP has been interpreted as evidence for polar body fusion following meiosis II, also known as automixis, leading to the restoration of diploidy in unfertilized eggs (*Figure 1A*; *Chapman et al., 2007*; *Card et al., 2021*; *Booth et al., 2023*; *Booth and Schuett, 2016*; *Reynolds et al., 2012*). If automixis involves the fusion of one of the meiotic products from the first polar body (central automixis), homozygosity will be concentrated near the chromosome ends and heterozygosity will be preferentially retained near the centromeres as premeiotic recombination strongly favors homologs over sister chromatids and homologs segregate during the first meiotic division. In contrast, a second polar body fusion (terminal automixis) would reunite sister chromatids, for which heterozygosity is preferentially seen near the chromosome termini (*Figure 1B*). In many cases, heterozygous and homozygous loci appeared to be inherited in FP

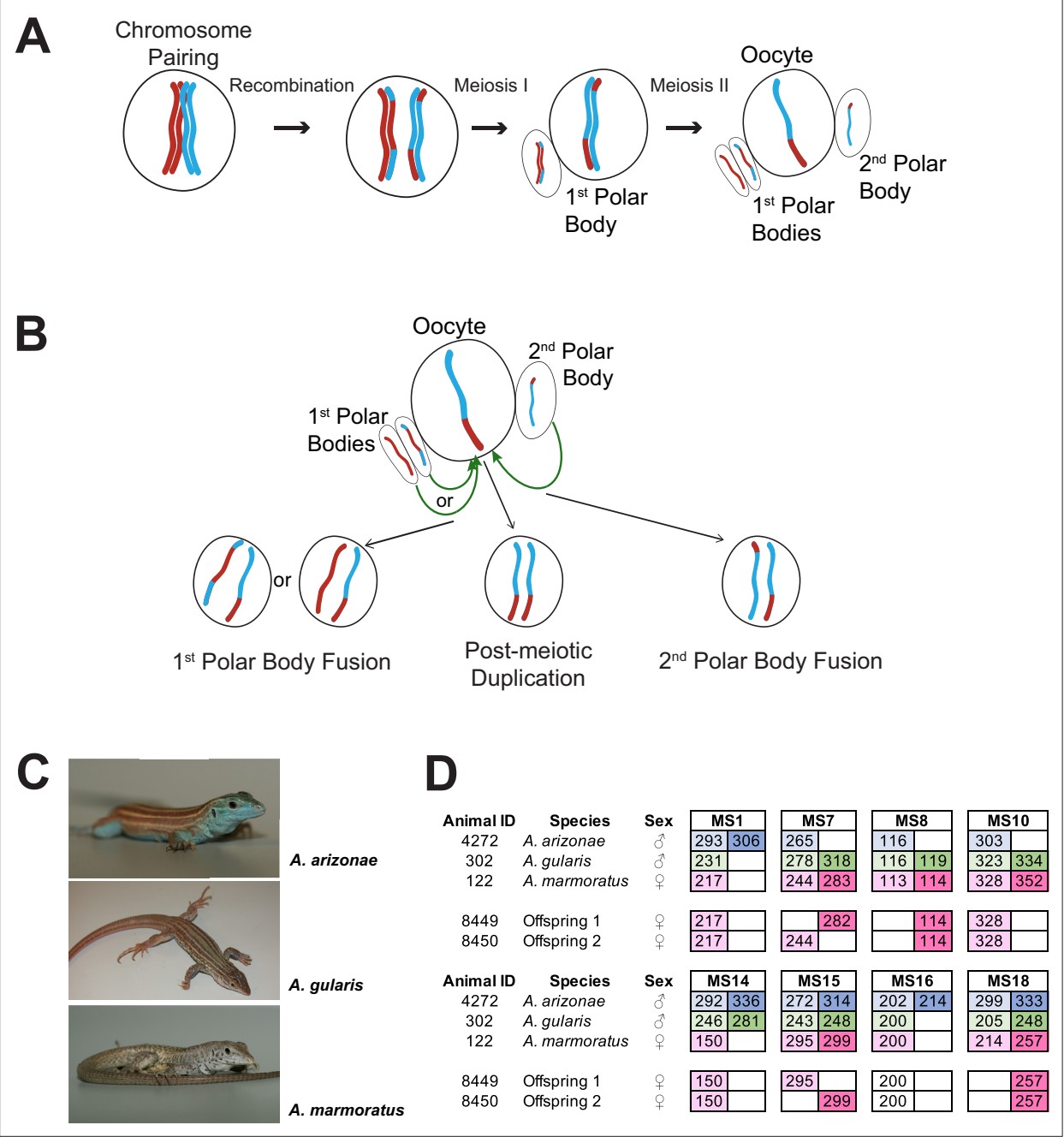

**Figure 1.** Overview. (**A**) Schematic of canonical meiosis. Only one pair of homologous chromosomes is shown using red and blue to distinguish homologs. (**B**) Schematic of main mechanisms by which a diploid oocyte may be produced in the context of facultative parthenogenesis. First polar body fusion, second polar body fusion, or post-meiotic duplication of chromosomes in the haploid gamete. (**C**) Photographs of *Aspidoscelis arizonae* with characteristic blue ventral coloration (top), *A. gularis* with light spots in dark fields that separate light stripes on dorsum (middle), and *A. marmoratus* with light and dark reticulated pattern on dorsum (bottom). (**D**) Microsatellite analysis for the three co-housed animals and two offspring (ID 8449 and 8450) produced in this enclosure. Alleles are color-coded as follows: *A. arizonae* male (blue), *A. gularis* male (green), and *A. marmoratus* female (red). Differences in shading highlight the two alleles at heterozygous loci. Both offspring are homozygous at all loci with most alleles matching only maternal alleles. For MS16, offspring alleles are not shaded because of this allele being shared between the mother and the *A. gularis* male. Single nucleotide differences in size are common binning artifacts and, therefore, are not scored as different alleles.

The online version of this article includes the following figure supplement(s) for figure 1:

**Figure supplement 1.** Microsatellite analysis of eight loci for the three female *Aspidoscelis marmoratus* within the enclosure and the two offspring hatched in January 2009.

offspring (*Groot et al., 2003*; *Allen et al., 2018*; *Card et al., 2021*; *Kratochvíl et al., 2020*; *Booth et al., 2023*), but information as to the genomic location of these loci has been lacking. Central and terminal automixis are also distinguished by the extent of heterozygosity, with lower levels observed in next-generation sequencing data in snakes and crocodiles suggesting terminal automixis as the likely mechanism (*Allen et al., 2018*; *Card et al., 2021*; *Booth et al., 2023*).

While facultative parthenogenesis occurs in a wide range of vertebrate species, true obligate parthenogenesis is limited to a few taxa of squamate reptiles including the North American whiptail lizards of the genus *Aspidoscelis* (*Avise, 2015*). Historic hybridization events between distinct gonochoristic species in this clade has given rise to numerous hybrid individuals with the ability to reproduce clonally as all female lineages (*Vanzolini, 1993*; *Reeder et al., 2002*). In contrast to the increased homozygosity associated with FP, obligate parthenogenetic species are characterized by the long-term preservation of the high degree of heterozygosity that had its origin in the lineage-founding cross-species hybridization events.

Our laboratory has a longstanding interest in the mechanism of obligate parthenogenesis in whiptail lizards (*Lutes et al., 2010*; *Newton et al., 2016*). In this context, we are maintaining and propagating individuals of several obligate parthenogenetic as well as gonochoristic species. MS analysis revealed over 20 incidences of FP in the marbled whiptail lizard, *A. marmoratus* and the Arizona striped whiptail, *A. arizonae.* (The taxonomy of the genus *Aspidoscelis* has undergone frequent revisions and the maternal ancestor of the obligate parthenogenetic species *A. neomexicanus* was formerly known as the subspecies *A. tigris marmoratus* and the male ancestor as the subspecies *A. inornatus arizonae* (*Barley et al., 2022a*). For the purpose of this manuscript we follow the taxonomic conclusions by *Barley et al., 2021*). Whiptail lizards have an XX/XY sex determination system (*Cole et al., 1969*) and all FP offspring are consequently female. The identification of multiple incidences of FP provided us with the opportunity to investigate the mechanism of FP in whiptail lizards through next-generation sequencing. The generation of a genome assembly, in addition to whole-genome sequencing, allowed us to distinguish between different mechanisms for restoring diploidy in FP animals. To address the question of whether FP is limited to animals in captivity, we examined reduced-representation sequencing (RAD-seq) data of 321 whiptail lizards from 15 gonochoristic species sampled in nature. In aggregate, a combination of MS analysis, next-generation sequencing, and cytological analysis allows us to report on both the evidence and mechanism of FP in whiptail lizards and suggest that a baseline incidence of FP may coexist alongside sexual reproduction in some species.

## Results
### Identification of FP in *A. marmoratus*

In the context of studying interspecific hybridization among gonochoristic species of whiptail lizards, three female *A. marmoratus* (ID 122, 4238, 4239) were housed with a male *A. arizonae* (ID 4272) and a male *A. gularis* (ID 302) for close to three years. During this period, seven hybrid offspring between *A. marmoratus* 122 and *A. arizonae* 4272 were produced and confirmed by MS analysis. These animals will be described in more detail in due course. Surprisingly, two female hatchlings emerged that resembled *A. marmoratus* rather than the expected products of hybridization with either *A. arizonae* or *A. gularis*. Genotyping revealed only a single allele for each of eight MS markers in the two offspring (ID 8449 and 8450, *Figure 1D*) and identified *A. marmoratus* (ID 122) as the mother (*Figure 1—figure supplement 1*). The mother and the male *A. arizonae* were each heterozygous at five of the eight markers and the *A. gularis* male at six. Further supporting a uniparental origin of 8449 and 8450, all alleles found in the offspring were also present in the mother (*Figure 1D*). For seven of the eight markers, neither male shared the allele found in the hatchling lizards, providing strong evidence that neither male fathered the offspring. For the remaining marker MS16, the *A. marmoratus* mother and the *A. gularis* male were homozygous for the same allele found in the two offspring, therefore, not allowing a conclusion to be based on this locus. It is important to note that for two of the markers (MS7 and MS15), the two offspring inherited different alleles from the mother, indicating that they are not genetically identical to each other, but have randomly inherited one of the maternal alleles at each locus.

Two additional eggs (ID 8394 and 9070) were recovered from the same enclosure and found to contain developing embryos. MS analysis also revealed *A. marmoratus* 122 as the mother and

complete homozygosity at all loci. Given that all of these offspring are female, inherited only maternal alleles, and animal 122 had no history of being housed with a conspecific male during its lifetime, both interspecific hybridization and long-term sperm storage are all but ruled out and FP is strongly supported.

FP animals of *A. marmoratus* presented a unique opportunity to examine the underlying molecular mechanism. The observed homozygosity at all MS loci further promised to aid in the generation of a high-quality genome assembly, as homozygosity circumvents the challenge of collapsing haplotypes into a consensus sequence (*Kajitani et al., 2014*). To increase homozygosity, inbreeding for 15–20 generations is common practice prior to whole-genome sequencing and genome assembly (*Zhang et al., 2019*; *Fang et al., 2012*). However, generation times of more than one year make this a costly and time-consuming strategy for many vertebrate species including *A. marmoratus*.

## Genome sequencing and de novo assembly

The *A. marmoratus* genome is distributed over 23 chromosomes as previously demonstrated by metaphase spread analysis (*Lowe and Wright, 1966*). We used flow cytometry to compare nuclear DNA content of *A. marmoratus* erythrocytes from whole blood with cells from three species with well-characterized genome sizes. The nuclear DNA content of *A. marmoratus* was close to that of *Danio rerio* (1.4 Gb) and we calculated a haploid genome size for *A. marmoratus* of 1.55 Gb (*Figure 2—figure supplement 1A*).

Genomic DNA of FP animal 8450 was used to generate short insert paired-end, mate-pair (5 Kb, 8 Kb, 2–15 Kb, 40 Kb), and Chicago (*Putnam et al., 2016*) libraries for Illumina short-read sequencing. The paired-end and mate-pair reads were first assembled with Meraculous (*Chapman et al., 2011*) yielding an N50 of 1.6 Mb. The subsequent addition of Chicago reads and scaffolding with the HiRise pipeline by Dovetail Genomics produced an assembly of 1,639,530,780 bp distributed over 3826 scaffolds (*Supplementary file 1*) and raised the scaffold N50 to 32.22 Mb (*Figure 2—figure supplement 1B*). With a BUSCO completeness score of 96% the *A. marmoratus* genome assembly is comparable to other recently released reptilian genome assemblies (*Figure 2—figure supplement 1C*). Over 98% of the assembled sequences are contained within 90 scaffolds of more than 1 Mb in length, making this assembly highly contiguous.

Phylogenetic analysis of shared BUSCO genes with several other reference genomes (*Xenopus*, zebrafish, medaka, platyfish, tegu, green anole, chicken, mouse, rat, dog, cow, human) confirmed that *A. marmoratus* is most closely related to the tegu *Salvator merianae*, another representative of the family Teiidae (*Figure 2—figure supplement 2*). As transposable elements are a driving force in genome evolution, we examined the repeat content for the *A. marmoratus* genome. All classes of repeat elements combined amounted to 40.27% of the *A. marmoratus* assembly, only slightly below that found in other lizards *S. merianae* and *Anolis carolinensis* (*Figure 2—figure supplement 3A*). Strikingly, unclassified repeats make up the largest class of repeat elements in the *A. marmoratus* genome, an observation that parallels findings in *S. merianae*. However, a comparison between the unclassified repeats found in *A. marmoratus*, *S. merianae*, and *A. carolinensis* revealed few similarities with only around 10% of the unclassified repeats shared between *A. marmoratus* and *S. merianae*, and no significant overlap between these two species and *A. carolinensis* (*Figure 2—figure supplement 3B*). While further characterization of the unclassified repeat elements is needed, it is apparent that an impressive expansion of novel repeat element classes has occurred within this clade.

To annotate the *A. marmoratus* genome, we assembled a total of 119,728 transcripts from RNA-seq data generated from blood and embryo using Trinity (*Supplementary file 2*). These transcripts were subsequently used in the MAKER2 gene annotation pipeline, yielding 25,856 protein-coding genes and 44,461 protein isoforms (*Supplementary file 3*). To assign a putative function, we used BLASTp to query the UniProtKB/Swiss-Prot database (*UniProt Consortium, 2018*) and found significant hits for 76% of the putative protein-coding genes. Our assembly and annotation pipelines yielded 40 *HOX* genes and 2 *EVX* genes in four gene clusters (*Figure 2—figure supplement 4*). The *HOX* gene clusters are highly conserved among tetrapods and their complete presence and shared order within each cluster serve as a measure of assembly quality (*Kuraku and Meyer, 2009*).

## Assessment of heterozygosity

The presence of only one allele for each of the examined MS markers already suggested widespread homozygosity in *A. marmoratus* produced by FP, consistent with similar observations in other vertebrate species. The highly contiguous genome assembly now afforded us the opportunity to probe the mechanism of FP by searching for regions of heterozygosity and mapping their relative genomic locations. Towards this aim, we performed whole-genome sequencing for an additional nine animals: four of them produced by FP (ID 12512, 12513, 6993, 9177), two mothers (ID 122, 9721), and three unrelated control animals (ID 003, 001, S30700; *Supplementary file 4*). Each mother and the controls were heterozygous at several MS markers confirming their origin through sexual reproduction. Following alignment to the reference genome (*Supplementary file 5*), we defined heterozygous sites as those covered by an even number of reads with two alleles supported by the same number of reads. Sites covered by an odd number of reads were filtered out for this initial analysis. This stringent requirement was chosen to limit the search to apparent heterozygous sites with strong support, decreasing the chance of false positives.

For all ten individuals, the average sequencing coverage ranged between 15.91 and 20.08 (*Figure 2—figure supplement 5*). For FP animals, the number of heterozygous sites in a 10 Kb sliding window approaches zero for all sites with mean coverage (*Figure 2A*). For positions with coverage greater than the average, an increase in apparent heterozygosity was observed, due to the collapse of repetitive sequences during the assembly process. Based on this observation, we limited further analysis to positions in the genome where the coverage is equal to the mean sequencing depth (as defined by rounding the mean sequencing coverage value to the next even integer). For example, for animal 003, the average sequencing coverage is 18.31 (*Figure 2—figure supplement 5A*) and we only considered sites with a coverage of 20 (*Figure 2—figure supplement 6A*). This generated between 30,769 and 53,416 heterozygous sites for which two alleles were equally supported in the sexually produced mothers and control animals (*Figure 2—figure supplement 6A–E*). Far fewer heterozygous sites (between 649 and 928) were observed in the FP animals (*Figure 2—figure supplement 6F–J*). Plotting the heterozygous sites according to their position in the reference genome illustrates not only their sparsity in the genomes of FP animals, but also reveals their random distribution (*Figure 2B*). If FP genomes were the product of automixis, regions of homozygosity would be interspersed with regions of heterozygosity. The extent of heterozygosity within the latter would be the same as that observed in the respective mother. The few apparently heterozygous sites identified in FP animals are, therefore, not supporting either form of automixis but are most likely the result of over-assembly of repetitive regions (ie. collapsing paralogous loci into a single representative sequence) and a combination of biological (ie. somatic mutations) and technical errors (ie. PCR and sequencing errors).

When examining each mother-daughter group, the average number of heterozygous sites per 10 Kb window was greater in the sexually produced mothers across the entire assembly (*Figure 2C*). For most of the assembly, the average number of heterozygous sites remained close to zero for FP animals. The most notable exception was a region around genomic coordinate 1.0 Gb, but even there the extent of apparent heterozygosity remained below 50% of what is observed in this region for each of the mothers. Examination of the scaffold in question (Scaffold 45) revealed 167 genes annotated as homologous to *vomeronasal 2 receptor 26* (*Vmn2r26*; *Figure 2—figure supplement 7*, *Figure 2—figure supplement 8*). Members of this subfamily of receptors are found on the microvilli of the vomeronasal organ, where they are responsible for pheromone detection and play a significant role in social and environmental responses (*Ryba and Tirindelli, 1997*; *Houck, 2009*; *Su et al., 2009*). Given that this genomic region harbors a large cluster of highly similar genes, the most parsimonious explanation for the elevated level of apparent heterozygosity is over-assembly. This conclusion is further supported by the increase in apparent heterozygosity in this region for the mothers and control animals. In aggregate, our analysis strongly supports genome-wide homozygosity for FP animals, inconsistent with either central or terminal automixis. Instead, the results favor a post-meiotic mechanism that restores diploidy by replicating the haploid genome residing in the oocyte following completion of the two meiotic divisions and thereby establishing genome-wide homozygosity in the offspring.

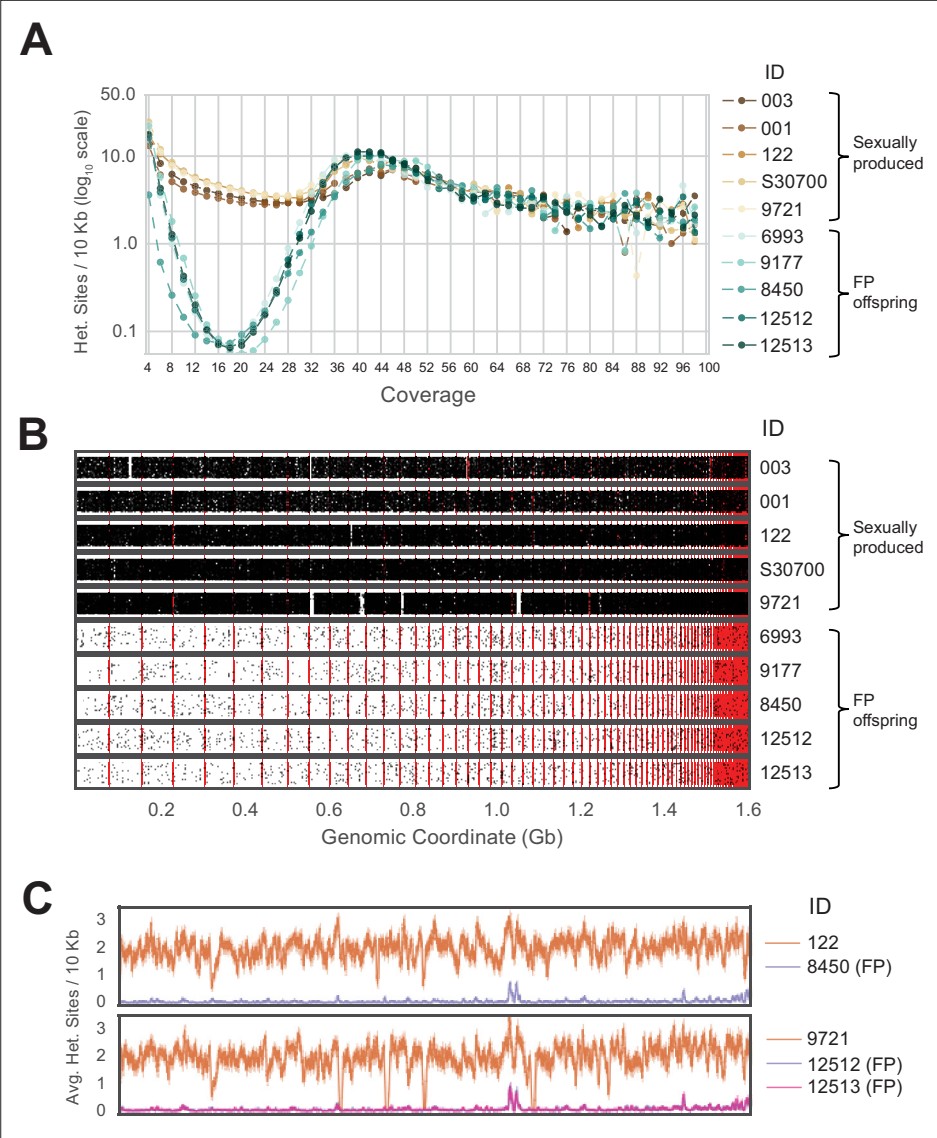

**Figure 2.** Genome-wide homozygosity in animals produced by facultative parthenogenesis. (**A**) Effect of coverage on the apparent rate of heterozygosity based on evenly split read counts supporting two alleles. Analysis of whole-genome sequencing data for five sexually produced animals (ID 003, 001, 122, S30700, 9721) and five individuals produced by facultative parthenogenetic (FP) animals (ID 6993, 9177, 8450, 12512, 12513) were aligned to the reference genome. In FP animals, the number of heterozygous sites approaches zero for sites with mean coverage ($\bar{x}$=18.37). (**B**) Scaffolds are ordered from largest to smallest on the x-axis. Red lines indicate borders between ordered scaffolds. Each black dot represents a heterozygous position in the genome defined by having a sequencing coverage equal to the average and equal support for only two alleles. The y-axis position of each data point is a random value between bounds of area shown to spread the data and better illustrate the density of heterozygous sites. (**C**) Average heterozygous sites, as defined in (**B**), per 10 Kb window for mothers (orange) and respective FP daughters (purple and pink).

The online version of this article includes the following figure supplement(s) for figure 2:

**Figure supplement 1.** Genome assembly of *Aspidoscelis marmoratus*.

**Figure supplement 2.** Maximum likelihood tree for 13 vertebrate genomes, based on 1333 single-copy BUSCOs detected across all species analyzed.

**Figure supplement 3.** Identification of unclassified repeat elements in the *A. marmoratus* genome.

**Figure supplement 4.** *Aspidoscelis marmoratus* HOX gene clusters.

**Figure supplement 5.** Distribution of sequence coverage across the genome for each animal.

*Figure 2 continued on next page*

*Figure 2 continued*

**Figure supplement 6.** Analysis of all positions in the genome at average coverage, for which reads support exactly two alleles.

**Figure supplement 7.** A 1Mb sliding window for Scaffold 45 showing the annotated vomeronasal 2 receptors homologs.

**Figure supplement 8.** Phylogenetic tree showing the evolutionary distance between mouse V2Rs (red branches) and *Aspidoscelis marmoratus* homologs (black branches).

## Cryptic FP in *A. arizonae*

Following the identification of several *A. marmoratus* generated by FP, we genotyped individuals from two other gonochoristic species housed in our laboratory. While no cases of FP were identified among 80 *A. gularis* produced eggs in captivity, we identified eight incidences of FP among 832 *A. arizonae* records between October 2007 and July 2018. During the same period, we recorded 15 incidences of FP among 286 *A. marmoratus* records (*Supplementary file 6*). Notably, in all cases, eggs undergoing FP development had been laid in enclosures where females were housed with conspecific males or males of a sister species known to mate with the heterospecific females. Isolation from mating partners was thus not a significant factor in triggering FP. In one enclosure, three *A. arizonae* females (ID 12850, 12851, 12852) were housed with a conspecific male (ID 12849; *Figure 3A*). MS analysis of four hatchlings that originated from a single clutch laid in this enclosure identified animal 12852 as the mother of all four animals. Unexpectedly, two of her offspring were homozygous at all eight loci examined, whereas the two others were heterozygous at all loci, identifying animal 12849 as their father (*Figure 3B*). Therefore, both fertilized and unfertilized eggs developed alongside each other within the same clutch.

## Mixoploid erythrocytes and developmental defects

Microscopic examination of blood from a newly hatched FP lizard revealed a striking bimodality in the sizes of erythrocyte nuclei when compared to a sexually produced animals (*Figure 4A, B*). Nuclear size correlates well with DNA content measurements (*Walker et al., 1991*), suggesting the presence of mixoploidy in the FP animal. Whereas most cells closely resembled those observed in the blood from the sexually produced animal, approximately 10% of red blood cells from the FP animal harbored smaller nuclei, consistent with half the amount of DNA (*Figure 4B*). In addition, 1.27% contained two small nuclei, indicating that the final cytokinesis during erythrocyte differentiation had failed for some haploid progenitor cells. DNA content analysis by flow cytometry confirmed the presence of haploid cells (*Figure 4C*). In the blood of sexually produced animals, no haploid or bi-nucleated cells were observed. These observations raise the possibility that the embryonic development of FP animals is initiated with consecutive divisions of a haploid, unfertilized oocyte. At a later stage in development, diploid cells most likely arise via failed cytokinesis. From that point forward, both haploid and diploid cells coexist, and the embryo develops in a mixoploid state. Indicative of a more widespread phenomenon, mixoploidy was also observed in another FP *A. marmoratus* and in *A. arizonae* (*Figure 4—figure supplement 1*). The observed fraction of haploid cells was closer to 1% in these instances.

Genome-wide loss of heterozygosity exposes functionally compromised alleles that were previously covered by intact alleles on the homologous chromosomes. Depending on the extent of this genetic load, one would expect a substantial fraction of oocytes to not develop at all or for defects to manifest at various stages of embryonic and post-embryonic development. Indeed, of the 23 incidences of FP examined here, only 14 hatched, while the remaining lizards died in ovum (*Supplementary file 6*). For these nine unhatched eggs, we isolated developed embryos shortly after the expected hatch date and confirmed FP origin by MS analysis. The clutches that produced the 23 confirmed cases of FP contained an additional 24 eggs. For these, development did not initiate or terminated at an earlier stage of development precluding MS analysis. Based on the uncertainty regarding how many of these eggs underwent partial FP development, the incidence of FP may be even higher than reported here.

The observation of various malformations in several of the FP embryos and hatchlings further supports that genome-wide homozygosity unmasks deleterious alleles. Notable developmental defects included craniofacial abnormalities such as misaligned jaws, agenesis of eyes, missing limbs, and failed abdominal closure (*Figure 4—figure supplement 2*). Only six out of 16 FP animals (37.5%)

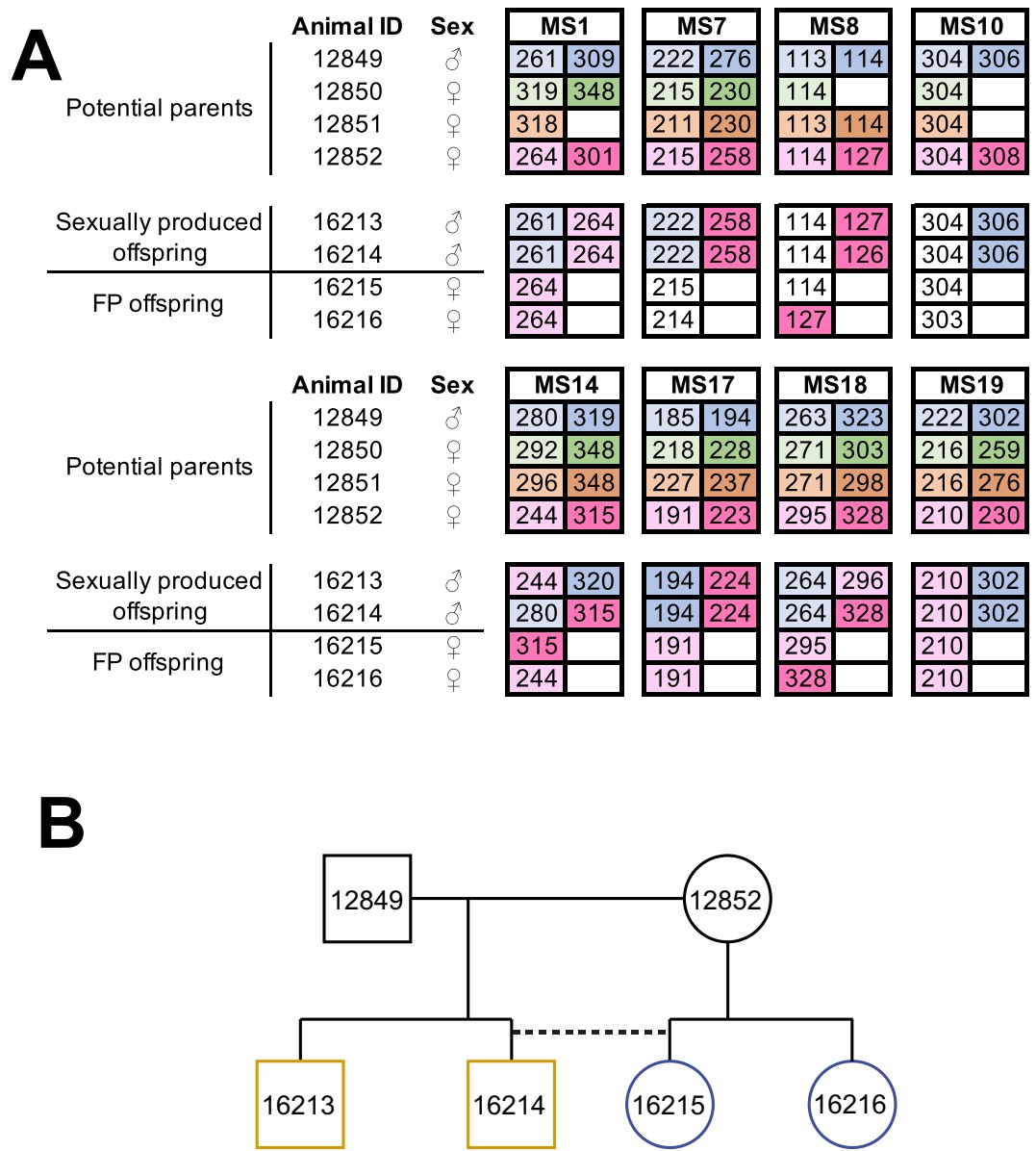

**Figure 3.** Facultative parthenogenesis is also found in *Aspidoscelis arizonae*. (**A**) Microsatellite analysis for the four co-housed adult animals (ID 12849, 12850, 12851, 12852) and the four hatchlings (ID 16213, 16214, 16215, 16216) produced in this enclosure. Alleles are color-coded for each potential parent: 12849 male (blue), 12850 female (green), 12851 female (orange), 12852 female (red). Differences in shading highlight the two alleles at heterozygous loci. Offspring 16213 and 16214 are heterozygous at all loci, with most loci having one allele matching 12849 and one allele matching 12852. Offspring 16215 and 16216 are homozygous at all loci, with most alleles matching only the 12852 female. Non-shaded offspring alleles indicate ambiguous inheritance as multiple adult animals share the same allele. Single nucleotide differences in size are common binning artifacts and, therefore, are not scored as different alleles. (**B**) Pedigree shows the relationship between the four offspring. The single clutch of four contains both sexually (yellow) and facultative parthenogenetically (blue) produced offspring.

hatched with no discernable developmental defects (*Figure 4—figure supplement 2 A-B*). This is in stark contrast to sexually produced animals, where over 98% of hatchlings (n=687) showed no abnormalities. Additionally, most of the defects noted in sexually produced animals were less severe than in FP animals including bulges in tails or truncated digits. While we have not recorded instances of FP animals producing offspring via FP, as described for the whitespotted bamboo shark (*Straube et al., 2016*), FP *A. marmoratus* 8450 did produce two eggs while housed in isolation, but these failed to hatch. Analysis of the ovaries of FP animal 8450 as well as germinal vesicles of its FP sister 8449

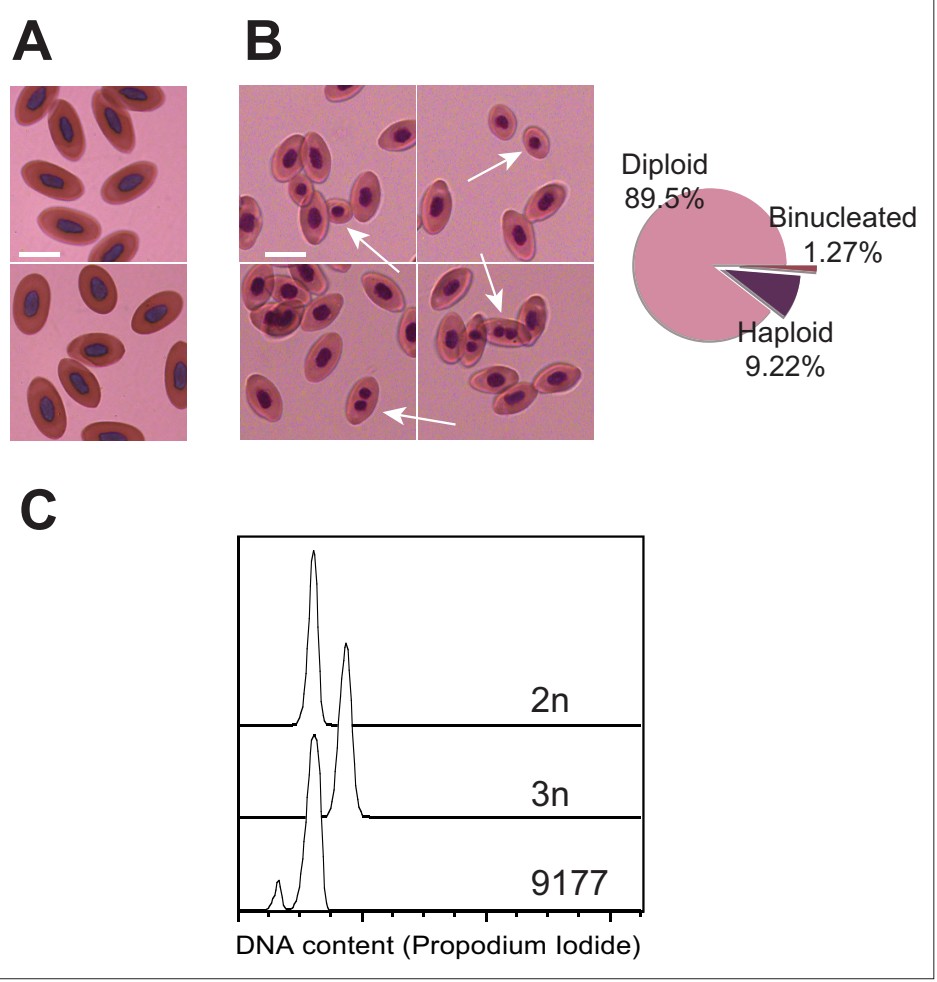

**Figure 4.** Detection of mixoploidy associated with facultative parthenogenesis. (**A**) Giemsa staining of erythrocytes from a sexually produced *Aspidoscelis marmoratus* (ID 14744). All cells are diploid (n=601). Scale bar corresponds to 10 μm. (**B**) Giemsa staining of erythrocytes from a newly hatched facultative parthenogenetic *A. marmoratus* (ID 9177). Diploid (n=844), smaller haploid (n=87), and binucleated (n=12) cells are evident. Scale bar corresponds to 10 μm. (**C**) DNA content from erythrocytes determined by propidium iodide staining and detection by flow cytometry. Samples are from a sexually produced *A. marmoratus* (2n, ID 5358), an obligate triploid parthenogen *A. exanguis* (3n, ID 4950), and facultative parthenogenesis (FP) *A. marmoratus* (ID 9177). Number of events scored by flow cytometry were 44,145 (2n), 44,043 (3n), and 44,060 (9177). The FP 9177 sample contained an additional peak to the left of the 2 n peak (90.04%), indicating the presence of haploid cells (9.62%).

The online version of this article includes the following figure supplement(s) for figure 4:

**Figure supplement 1.** Mixoploidy detected in both *A. marmoratus* and *A. arizonae*.

**Figure supplement 2.** Animals produced by facultative parthenogenesis.

**Figure supplement 3.** Ovaries of *Aspidoscelis marmoratus* facultative parthenogenesis (FP) animal 8450 and germinal vesicles of FP sister 8449 revealed no differences in structure and anatomy compared to fertile sexually reproducing animals.

revealed no differences in structure and anatomy compared to fertile sexually reproducing animals (*Figure 4—figure supplement 3*).

## Putative incidences of FP in wild populations of whiptail lizards

As FP has been associated with captivity in most species where it has been reported, we examined restriction site-associated DNA sequencing (RAD-seq) data for wild animals across 15 gonochoristic species (*Figure 5*). Because RAD-seq is a form of reduced-representation sequencing (*Rivera-Colón and Catchen, 2022*), we limited further analysis to the 321 individuals that had an average sequencing

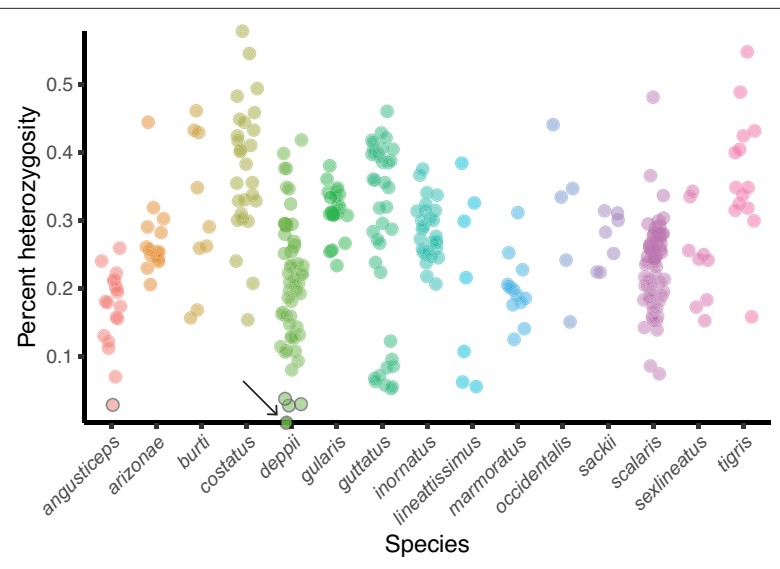

**Figure 5.** Heterozygosity estimates of whiptail lizards collected in nature. Percent heterozygosity estimates from reduced-representation sequencing (RAD-seq) for 321 whiptail lizards from 15 species. All individuals had an average coverage of at least 20. Each point is an individual, and percent heterozygosity was calculated only for sites where the coverage is equal to the average sequencing coverage. Five points with black borders indicate individuals (one *angusticeps* and four *deppii*) with low levels of heterozygosity. The heterozygosity of *Aspidoscelis deppii* ID LDOR30 (marked with arrow) is far less than that observed for individuals of the same species (Rosner's Test for Outliers within *deppii* individuals, p<0.001), having only one called heterozygous position. Species (sample size): *angusticeps* (**Booth et al., 2023**), *arizonae* (**Lenk et al., 2005**), *burti* (**Germano and Smith, 2010**), *costatus* (**Shibata et al., 2017**), *deppii* (**Ryba and Tirindelli, 1997**), *gularis* (**Olsen and Marsden, 1954**), *guttatus* (**Reeder et al., 2002**), *inornatus* (**Ryder et al., 2021**), *lineattissimus* (**Groot et al., 2003**), *marmoratus* (**Allen et al., 2018**), *occidentalis* (**Dudgeon et al., 2017**), *sackii* (**Groot et al., 2003**), *scalaris* (**Streisinger et al., 1981**), *sexlineatus* (**Germano and Smith, 2010**), *tigris* (**Card et al., 2021**).

The online version of this article includes the following source data for figure 5:

**Source data 1.** Source data of *Figure 5*.

coverage of at least 20. Computational data analysis revealed five animals (one *A. angusticeps* and four *A. deppii*) that had very low levels of heterozygosity at positions of average coverage (<0.05% heterozygous positions). In contrast, the average level of heterozygosity was 0.261% across the dataset, with the highest heterozygosity value at 0.578%. Of the five low heterozygosity animals, *A. deppii* (ID LDOR30) showed the most striking level of homozygosity affecting all sites but one (Rosner's Test for Outliers within *A. deppii* individuals, log-transformation, $R$=5.127, $\lambda$=3.928, p<0.001). This is consistent with the pattern of homozygosity observed with the whole-genome sequencing from FP animals produced in the laboratory. Further fieldwork and analysis will be required to assess the level of FP in natural populations of gonochoristic *Aspidoscelis* species (and other factors that could influence the observed heterozygosity such as population size, levels of hybridization, and inbreeding). To assess whether all whiptail species can produce viable offspring through FP, larger and broader datasets will be required to compare the incidence of FP between species, especially because animals with developmental defects associated with FP would not have hatched or survived in the wild and would, therefore, not have shown up in the RAD-seq dataset.

## Discussion

In this study, we report over 20 incidences of facultative parthenogenesis in marbled and Arizona striped whiptail lizards. By sequencing and assembling a highly contiguous *A. marmoratus* genome, we were able to refute automixis as the underlying mechanism in whiptail lizards. Instead, genome-wide homozygosity raises a possibility of a post-meiotic mechanism involving the activation of embryonic development in unfertilized haploid oocytes. Even though FP whiptail lizards are largely

comprised of homozygous diploid cells, a fraction of haploid cells persists through development and is readily detectable in young adults. Such mixoploidy and genome-wide homozygosity come at a price. In eight confirmed and 24 suspected cases of FP, development ceased prior to hatching and most FP animals that hatched showed congenital defects. Nevertheless, FP was observed at a rate of 1% and 5% in *A. arizonae* and *A. marmoratus*, respectively, and this occurred in the presence of mating partners. Interestingly, these rates are similar to what has been reported for wild populations of two North American pitviper species (*Booth et al., 2012*). Our findings indicate that FP is far more common in some vertebrate species than previously thought. The purifying selection associated with homozygosity may be an important force in generating additional resilience to counteract the effects of population bottlenecks and inbreeding depression. However, support for this hypothesis is predicated on the fitness and reproduction of FP offspring and, therefore, more long-term studies on seemingly healthy individuals of FP origin are needed.

The start of embryonic development is tightly coupled to fertilization in many vertebrates as the sperm entering the oocyte triggers a signaling cascade that is essential for the completion of female meiosis and initiation of cell division following karyogamy (*Sagata, 1996*). This process has been mimicked in the laboratory by piercing frog oocytes with a needle to trigger the signal to complete meiosis and initiate replication and division in the haploid oocyte (*Le Peuch et al., 1985*; *Wolf, 1974*). In zebrafish, homozygous embryos are routinely generated by fertilization of oocytes with UV-irradiated sperm, a treatment that destroys the paternal DNA (*Streisinger et al., 1981*). An oocyte treated in this manner will replicate the intact maternal genome in the absence of karyogamy. If the haploid oocyte is then subjected to heat shock treatment, cytokinesis is prevented resulting in a pseudodiploid oocyte that undergoes a second round of DNA replication followed by mitosis. Thus an entirely homozygous diploid embryo starts to develop (*Kroeger et al., 2014*). In contrast to human intervention forcing two consecutive rounds of DNA replication to occur without intervening mitosis at the start of embryonic development, our data indicate that one or multiple rounds of DNA replication and mitosis take place in some haploid oocytes of whiptail lizards prior to a skipped mitosis yielding a homozygous diploid cell followed by mixoploid development. Initiation of development in a haploid state may be conserved in avian species as unfertilized turkey eggs can yield embryos that contain 40% of haploid cells at blastoderm followed by a reduction to 1.3% within the blood of hatched birds (*Cassar et al., 1998*). Depending on the tissue and ploidy distribution, the presence of haploid cells may contribute to abnormal development of specific tissues reported here and elsewhere (*Ito et al., 1991*; *Tanaka et al., 2004*). Successful embryonic development from haploid cells that restore the diploid state by duplication has also been observed in a stick insect species (*Pijnacker, 1969*). In whiptail lizards, we have not been able to examine post-meiotic oocytes as locating the post-meiotic nucleus within a large yolked egg is inherently difficult. The difficulty is compounded by the unpredictability of which eggs will undergo FP development and the need to sacrifice animals to remove eggs.

In addition to mixoploidy, genome-wide homozygosity constitutes another obstacle to normal development as each recessive deleterious allele is exposed in either the hemizygous state (haploid cells) or homozygous state (diploid cells). Indeed, arrested development and abnormal phenotypes are observed in FP whiptails, as well as in FP animals across many other species (*Booth et al., 2023*; *Booth and Schuett, 2016*; *Adams et al., 2023*; *Olsen, 1973*). It is important to note though that some whiptails of FP origin developed normally, much like their sexually produced counterparts. At the population level, FP leads to a precipitous reduction in genetic diversity as only one set of alleles is inherited in the next generation. While FP could be an adaptive trait to bridge population bottlenecks when mate encounters are infrequent, small populations already rely heavily on inbreeding and FP further reduces the size of the gene pool (*Watts et al., 2006*; *Ryder et al., 2021*).

While FP can be considered the most extreme example of inbreeding, it is also the most powerful example of genetic purging as it eliminates most deleterious alleles in a single generation. FP in whiptail lizards and other species could, therefore, be considered a reproductive strategy, akin to mixed-mating systems in plants (*Goodwillie et al., 2005*). The ability to produce offspring via two different strategies provides a level of reproductive assurance in plants (*Busch and Delph, 2012*). Indeed, we see parallels to this in our own data in which female whiptails have produced offspring via both sexual reproduction and FP on separate occasions or simultaneously within a single clutch. Within plant species with mixed-mating, there are differences between the rates of selfing and outbreeding between populations, and hypotheses as to why these differences occur include limited pollinator

visitation and resource availability (*Whitehead et al., 2018*). To assess whether the co-occurrence of sexual and FP reproduction in vertebrates can indeed be considered a reproductive strategy rather than biological noise will require further studies to assess the reproductive competence and fecundity of offspring produced by either mode of reproduction. To gain a better understanding of the origin and outcomes of FP in whiptail lizards, it will also be important to identify the triggers. It has been proposed that lack of or limited mate encounters triggers FP, but our data in combination with many other reports (*Booth et al., 2012*; *Booth et al., 2014*; *Kratochvíl et al., 2020*; *Ryder et al., 2021*; *Booth et al., 2011*; *Feldheim et al., 2023*; *Larose et al., 2023*) rejects the idea that this is the key trigger. Recent work identifying key cell cycle genes inducing FP in two species of *Drosophila* (*Sperling et al., 2023*) and selection resulting in higher incidences of parthenogenesis in birds (*Parker et al., 2010*; *Olsen, 1975*; *Olsen et al., 1968*) suggest a genetic basis for the initiation of FP.

Our study adds two species of whiptail lizards to a growing list of vertebrates capable of FP and establishes that it occurs alongside sexual reproduction in the presence of males. Using whole-genome sequencing, we demonstrate that post-meiotic genome duplication is the underlying mechanism. One must now consider the possibility that FP is an adaptive trait and that low rates of successful FP could contribute significantly to genome purification. Sexually mature FP offspring will have a low genetic load and only pass on neutral or mildly deleterious alleles to the next generation. However, a role for FP as an adaptive trait hinges on further studies demonstrating the ability of parthenogens to reproduce themselves either through further FP or sexually. If successive reproduction occurs, FP may reduce the frequency of deleterious alleles within a population, as well as provide reproductive assurance when males are scarce. Additional whole-genome sequencing data for species with documented FP will be needed for a better understanding of the genetic basis, propensity, and evolutionary significance of FP.

# Materials and methods

**Key resources table**

| Reagent type (species) or resource | Designation | Source or reference | Identifiers | Additional information |
|---|---|---|---|---|
| Biological sample (*Aspidoscelis spp.*) | Erythrocytes, tail clippings, liver isolation | This paper, wild populations | | See Ethics Statement |
| Chemical compound, drug | Giemsa stain | Sigma | GS500 | 0.40% |
| Chemical compound, drug | Schiff's reagent | Fisher Scientific | #SS32-500 | |
| Commercial assay or kit | Roche gDNA Isolation Kit | Roche | #11814770001 | |
| Commercial assay or kit | KAPA HTP kit | KAPA | KK8234 | |
| Commercial assay or kit | Nextera Mate-Pair Library Prep Kit | Illumina | FC-132–1001 | |
| Commercial assay or kit | TruSeq RNA Library Prep Kit v2 | Illumina | RS-122–2001 | |
| Sequence-based reagent | Oligos for MS analysis | This paper | | See *Supplementary file 7* |
| Software, algorithm | Meraculous 2.0 | https://jgi.doe.gov/data-and-tools/software-tools/meraculous/ | RRID:SCR_010700 | |
| Software, algorithm | BUSCO 3.0.1 | https://busco.ezlab.org/ | RRID:SCR_015008 | |
| Software, algorithm | RAxML 8.2.11 | https://cme.h-its.org/exelixis/web/software/raxml/ | RRID:SCR_006086 | |
| Software, algorithm | RepeatModeler 1.0.11 | https://www.repeatmasker.org/RepeatModeler/ | RRID:SCR_015027 | |
| Software, algorithm | RepeatMasker 4.0.9 | https://www.repeatmasker.org/ | RRID:SCR_012954 | |
| Software, algorithm | BLAST 2.6.0 & 2.9.0+ | https://blast.ncbi.nlm.nih.gov/Blast.cgi | RRID:SCR_004870 | |
| Software, algorithm | BWA 0.7.15 | https://bio-bwa.sourceforge.net/ | RRID:SCR_010910 | |

*Continued on next page*

*Continued*

| Reagent type (species) or resource | Designation | Source or reference | Identifiers | Additional information |
|---|---|---|---|---|
| Software, algorithm | Picard 1.119 | https://broadinstitute.github.io/picard/ | RRID:SCR_006525 | |
| Software, algorithm | GATK 3.5 | https://gatk.broadinstitute.org/hc/en-us | RRID:SCR_001876 | |
| Software, algorithm | seqtk 1.2-r94 | https://github.com/lh3/seqtk *Li, 2016* | 1.2-r94 | |
| Software, algorithm | pysam 0.12.0.1 | https://github.com/pysam-developers/pysam *pysam-developers, 2017* | 0.12.0.1 | |
| Software, algorithm | pysamstats 0.24.3 | https://github.com/alimanfoo/pysamstats *Miles, 2015* | 0.24.3 | |
| Software, algorithm | Trinity | https://github.com/trinityrnaseq/trinityrnaseq *trinityrnaseq, 2015* | 2.0.6 | |
| Software, algorithm | seqclean | https://sourceforge.net/projects/seqclean | | |
| Software, algorithm | MAKER2 2.31.8 | https://www.yandell-lab.org/software/maker.html | RRID:SCR_005309 | |
| Software, algorithm | Interproscan 5.13–52.0 | https://interproscan-docs.readthedocs.io/en/latest/ | RRID:SCR_005829 | |
| Software, algorithm | Exonerate 2.4.0 | https://www.ebi.ac.uk/about/vertebrate-genomics/software/exonerate | RRID:SCR_016088 | |
| Software, algorithm | Geneious 10.1.3 | https://www.geneious.com/ | RRID:SCR_010519 | |
| Software, algorithm | Stacks 2.62 | https://catchenlab.life.illinois.edu/stacks/ | RRID:SCR_003184 | |
| Software, algorithm | Micromanager 1.4 | https://micro-manager.org/ | RRID:SCR_000415 | |
| Software, algorithm | Flowjo treestar | https://www.flowjo.com/ | RRID:SCR_008520 | |

## Microsatellite analysis

DNA was extracted from tail samples for microsatellite genotyping as described in *Lutes et al., 2011*. PCR products were analyzed by capillary electrophoresis on a 3730 DNA Analyzer and data was analyzed using GeneMapper (v. 4.0). Primer information can be found in *Supplementary file 7*.

## Genome size estimation

The genome size of *A. marmoratus* was estimated by fluorescence-activated cell sorting (FACs), in which a standard curve correlating fluorescence intensity of DNA-bound propidium iodide with known genome sizes was generated using cells from fruit flies, zebrafish, and mouse, and then comparing fluorescent intensity with that of erythrocytes from *A. marmoratus*. Samples were stained using the Sigma PI staining preparation and analyzed on the Influx cytometer. PI fluorescence was collected using the PI Texas red detector with linear amplification and data analysis was performed in FlowJo and Microsoft Excel.

## DNA isolation, sequencing, and genome assembly for *A. marmoratus*

All genome sequencing libraries generated for the purpose of the *A. marmoratus* genome assembly were derived from the FP animal 8450. The liver tissue was first dissociated in a 10 mL Dounce

homogenizer using the tight-fitting pestle and then processed using the Roche gDNA Isolation Kit (#11814770001, MilliporeSigma, St. Louis, MO, USA).

A short insert, high-coverage library was generated using the KAPA HTP kit (KK8234), with 1 µg of gDNA. The resulting library was size selected for fragments between 500–850 bp on a Pippin Prep (Sage Science). Two 40 Kb mate-pair libraries were generated by Lucigen from 1 µg of gDNA using the CviQl and Bfal restriction enzymes, respectively. Each library was sequenced on the Illumina MiSeq using the MiSeq Reagent Kit v2 (500 cycles). An additional three mate-pair libraries were generated, spanning distances of 5 Kb, 8 Kb, and 2–15 Kb, using the Illumina Nextera Mate-Pair Library Prep Kit and 1 µg of gDNA for each. Size selection used the Gel-Plus protocol with Pippin for the 5 and 8 Kb libraries, and the Gel-Free protocol for the 2–15 Kb library. All three libraries were pooled and sequenced on three separate RapidSeq flow cells on an Illumina HiSeq 2500. Chicago libraries were prepared at Dovetail Genomics LLC, Santa Cruz, CA, USA from liver tissue to generate read pairs spanning distances up to 140 Kb and sequenced on an Illumina HiSeq 2500. The combined sequencing data was initially assembled at Dovetail Genomics using Meraculous and their in-house HiRise genome assembly algorithms to generate the *A. marmoratus* reference genome (AspMar1.0).

## Assessing assembly completeness

In order to assess the completeness of the *A. marmoratus* reference genome, we used BUSCO (v. 3.0.1) (*Simão et al., 2015*) with the vertebrate_od9 dataset containing 2586 genes, with default parameters apart from changing the BLAST cutoff from 1e-3 to a more stringent value of 1e-6. We used BUSCO numbers generated in *Gao et al., 2017* for *Shinisaurus crocodilurus* and *Alligator mississippiensis* in *Figure 2—figure supplement 1*.

To perform a phylogenetic analysis, 1333 shared 'complete' single-copy orthologs were identified in the genomes of green anole (*Anolis carolinensis*, anoCar2), cow (*Bos taurus*, ARS-UCD1.2), dog (*Canis lupus familiaris*, CanFam3.1), zebrafish (*Danio rerio*, danRer10), chicken (*Gallus gallus*, galGal5), human (*Homo sapiens*, GRCh38.p13), mouse (*Mus musculus*, GRCm38.p6), medaka (*Oryzias latipes*, oryLat2), rat (*Rattus norvegicus*, Rnor_6.0), Argentine black and white tegu (*Salvator merianae*, HLtupMer3), western clawed frog (*Xenopus tropicalis*, Xenopus_tropicalis_v9.1), platyfish (*Xiphophorus maculatus*, X_maculatus-5.0-male). For each amino acid sequence, a multiple sequence alignment was performed with MAFFT (v. 7.305) (*Katoh and Standley, 2013*). The alignments were concatenated into a supermatrix of 1,112,277 amino acids. Phylogenetic tree topology was estimated using the Maximum Likelihood inference method using the pthreads version of RAxML (v. 8.2.11) and the PROTOGAMMAAUTO model for sequence evolution with 100 bootstrap replicates (*Stamatakis, 2014*).

## Repeat identification

We quantified and annotated the repetitive DNA content within the *A. marmoratus* genome assembly by using the RepeatMasker pipeline on *A. marmoratus* scaffolds greater than or equal to 10 Kb in length. We first generated a de novo list of *A. marmoratus* repetitive elements using RepeatModeler (v. 1.0.11) (*Smit and Hubley, 2008*). We then used these as input into RepeatMasker (v. 4.0.9) (*Smit et al., 2008*) using the NCBI/RMBLAST (v. 2.6.0+) search engine. Unclassified repeat element consensus sequences from the RepeatModeler output for each of the three lizards (*A. marmoratus, S. merianae,* and *A. carolinensis*) were compared to each other by identifying reciprocal best hits using BLAST (v. 2.9.0+).

## Whole-genome sequencing, reference genome alignment, and heterozygosity determination

Genomic DNA isolated from either liver or tail was prepared for sequencing using the KAPA HTP Library Preparation Kit (KK8234). Stock adapters were used from the Nextflex kit and barcodes were from BioScientific. All libraries were sequenced on the Illumina HiSeq 2500 platform. Whole-genome sequencing data was aligned to the *A. marmoratus* reference genome with BWA (v. 0.7.15) (*Li and Durbin, 2010*) and marked for duplicates with Picard (RRID:SCR_006525; v. 1.119; https://broadinstitute.github.io/picard/). Because samples were sequenced over multiple lanes, the alignment files were merged subsequently, and another round of duplication marking was performed. The alignment files were realigned around small insertions and deletions with GATK (v. 3.5) (*DePristo et al., 2011*). Data

corresponding to lizard ID 122's bam file was down-sampled to 33% of its original size using seqtk (v. 1.2-r94) to match the expected average genome coverage of the other samples, as this animal was sequenced on one flow cell without multiplexing and, therefore, having much more sequenced reads.

The per position nucleotide profiles for each alignment were then generated using a combination of pysam (RRID:SCR_021017, v. 0.12.0.1) (https://github.com/pysam-developers/pysam) and pysam-stats (*Miles, 2015*. v. 0.24.3) (https://github.com/alimanfoo/pysamstats) to determine the heterozygosity at any genomic position.

## Transcriptome assembly and genome annotation

Two poly-A selected stranded RNA-sequencing libraries were generated with the TruSeq RNA Library Prep Kit v2 (RS-122–2001) and sequenced on an Illumina HiSeq 2500 for the purpose of an *A. marmoratus* transcriptome assembly. The first library was derived from a blood sample taken from a male animal, and the second library was derived from an embryo incubated at 28 °C and harvested 47–51 days post-egg deposition.

Trinity (v. 2.0.6) (*Grabherr et al., 2011*) was then used to generate an initial transcriptome assembly. The original reads were aligned to this transcriptome assembly using the Trinity companion script align_and_estimate_abundance.pl. Transcript isoforms with no read support were then filtered out and the remaining assembly was run through seqclean (https://sourceforge.net/projects/seqclean/). Evidence-based annotations for the transcriptome assembly were generated using the MAKER2 pipeline (v. 2.31.8) (*Holt and Yandell, 2011*). For MAKER2, the entire UniProtKB/Swiss-Prot database of proteins (*UniProt Consortium, 2018*) was used and the Repbase data base was used to mask repeats within the MAKER2 framework (*Bao et al., 2015*). Assigning putative functions to the gene models was performed using BLAST (v. 2.6.0) and Interproscan (v. 5.13–52.0) (*Jones et al., 2014*).

Copy number estimation for the Vomeronasal 2 receptor 26 (Vmn2r26) genes was based on aligning the mouse ortholog (http://www.orthodb.org), to the *A. marmoratus* reference assembly using Exonerate (v. 2.4.0) (*Slater and Birney, 2005*) with a maximum intron size set to 20 Kb. Genes annotated as Vmn2r26 in the MAKER2 annotations were concatenated and aligned using Geneious (v. 10.1.3) (*Geneious Prime, 2017*) with default settings. The FastTree plugin (v. 1.0) was used to generate the phylogenetic tree from the alignment with default parameters.

## RAD-sequencing analysis

Double digest RAD-sequencing data was derived primarily from previous studies (*Barley et al., 2021*; *Barley et al., 2022b*; *Barley et al., 2019*; *Barley et al., 2022a*). Fastq files were processed with Stacks (v. 2.62) using process_radtags and then samples from the same species were passed separately using denovo_map.pl. The script executes all the components of the standard Stacks pipeline. The gstacks program of denovo_map.pl calls variants for each position in each locus and assigns it either homozygous, heterozygous, or unknown. Output files were used to calculate the average coverage for each sample. Samples that have at least a coverage of 20 were considered for subsequent analysis. For each passing sample, the heterozygosity was calculated for sites at average coverage by adding up all heterozygous positions and dividing it by the total.

## Giemsa staining of erythrocytes

Whole blood was collected aseptically from tails using acid citrate dextrose as an anticoagulant. All samples were prepared immediately after collection. A 5 µL aliquot of the diluted blood sample was used to prepare a blood smear on a 25 × 75 × 1 mm microscope slide. Once dry, the slide was placed in 95% ethanol for 5 min. Giemsa stain (0.4%; Sigma, GS500) was applied liberally to cover the slide every 4 min for a total incubation time of 16 min. The prepared slides were imaged on an Axioplan2 imaging microscope equipped with a plan-apochromat 100 x /1.40 Oil objective and an Axiocam HRc (color) camera (Zeiss). Micromanager (v. 1.4) software was used to acquire the images. The acquired images were then scored visually for the number of haploid, diploid, and binucleated erythrocytes.

## Feulgen staining of erythrocytes

Whole blood was collected aseptically from tails using acid citrate dextrose as an anticoagulant. All samples were prepared immediately after collection. A 5 µL aliquot of the diluted blood sample was used to prepare a blood smear on a 25 × 75 × 1 mm microscope slide. Blood smears were treated

with 10% neutral buffered formalin for 5 min at RT, then rinsed twice in distilled water. The slides were immersed in 5 M HCl for 30 min at RT, and then rinsed 2 times in distilled water. Slides were then immersed in Schiff's reagent (Fisher Scientific #SS32-500) at RT for 15–30 min until nuclei were stained. The slides were transferred directly to bisulfite water that was prepared by dissolving 2.5 g of potassium metabisulfite in 500 mL of water and adjusting the pH to 4.0 by the addition of concentrated HCl. The bisulfite wash was repeated three times with 10–15 sec of agitation. The slide was then washed under running tap water for 2 min and dehydrated by incubating in 70% EtOH for 5 min and then 95% EtOH for 5 min. The preparations were cleared in xylene before mounting.

## Flow cytometry of erythrocytes

Blood collected from animals was treated as previously described for flow cytometry with modifications: ethanol fixation was performed after RNase treatment and propidium iodide staining was performed overnight, followed by sonication to disrupt aggregate cells (*Lutes et al., 2011*). A minimum number of 50,000 events were collected for each sample. Flowjo (treestar) was used for data analysis.

## Imaging of ovaries and germinal vesicle

Germinal vesicle isolation and acquisition of image stacks by confocal microscopy were performed as described (*Lutes et al., 2010*). Ovary images were acquired with a Leica M205FA dissection microscope with a planar 0.63 X objective using Micromanager (v. 1.4) software.

## Acknowledgements

We are grateful for the husbandry staff at the Stowers Institute for Medical Research (SIMR) especially Rick Kupronis and the team of dedicated reptile technicians, David Jewell, Alex Muensch, Jillian Schieszer, Christina Piraquive, and Kristy Winter, for outstanding husbandry and herpetocultural skills. We thank the members of the New Mexico Department of Game and Fish and the Arizona Game and Fish Department for their support. We thank Veronica Cloud for help with RNA extractions and the molecular biology, flow cytometry, and advanced microscopy core facilities at SIMR for their excellent technical support and the members of the Emergent AI Center at Johannes Gutenberg University Mainz (JGU) for outstanding IT support and helpful discussions. We acknowledge computing time granted on the supercomputer MOGON II at JGU as part of NHR South-West (nhrsw.de), as well as the Institute of Molecular Biology Bioinformatics Core for computing resources. This work was funded in part by the Howard Hughes Medical Institute, SIMR, and an Alexander von Humboldt Professorship awarded to P.B. at JGU. This work also received support from the National Science Foundation grant DEB-1754350 and the GenEvo RTG funded by the Deutsche Forschungsgemeinschaft (German Research Foundation) – GRK2526/1 – Projectnr. 40.7023052 and the Institute for Quantitative and Computational Biosciences (IQCB) in Mainz. We thank Charles (Jay) Cole, Uri García-Vázquez, Rick Kupronis, Daniel Lara-Tufiño, Norma Manríquez-Moran, Maximiliano Monroy, Priscilla Neaves, Adrián Nieto Montes de Oca, Harry Taylor, Robert Thomson, and Carol Townsend for assistance with fieldwork and specimen collection. We also thank the Natural History Museum of Los Angeles County, the UTEP Biodiversity Collections, Jay Cole and the American Museum of Natural History, the Museum of Vertebrate Zoology, Randy Klabacka and the Auburn University Museum of Natural History, and the LSU Museum of Natural Science for providing tissue loans in support of this research.

## Additional information

### Funding

| Funder | Grant reference number | Author |
|---|---|---|
| Stowers Institute for Medical Research | | Duncan Tormey |
| Howard Hughes Medical Institute | | Peter Baumann |

| Funder | Grant reference number | Author |
|---|---|---|
| Alexander von Humboldt-Stiftung | | Peter Baumann |
| Deutsche Forschungsgemeinschaft | GRK2526/1 | David V Ho |
| National Science Foundation | DEB-1754350 | Anthony J Barley |

The funders had no role in study design, data collection and interpretation, or the decision to submit the work for publication.

## Author contributions

David V Ho, Data curation, Software, Formal analysis, Validation, Investigation, Visualization, Writing – original draft, Writing – review and editing; Duncan Tormey, Data curation, Software, Investigation, Methodology, Writing – original draft; Aaron Odell, Formal analysis, Validation, Investigation, Visualization, Methodology; Aracely A Newton, Robert R Schnittker, Formal analysis, Investigation, Methodology; Diana P Baumann, Resources, Data curation; William B Neaves, Investigation; Morgan R Schroeder, Rutendo F Sigauke, Software, Formal analysis, Investigation, Methodology; Anthony J Barley, Resources, Investigation, Writing – review and editing; Peter Baumann, Conceptualization, Resources, Supervision, Funding acquisition, Validation, Writing – original draft, Project administration, Writing – review and editing

## Author ORCIDs

David V Ho http://orcid.org/0000-0002-4709-269X
Duncan Tormey https://orcid.org/0000-0002-0963-3424
Robert R Schnittker http://orcid.org/0000-0002-2950-8604
Anthony J Barley http://orcid.org/0000-0003-1675-6577
Peter Baumann http://orcid.org/0000-0003-4892-1485

## Ethics

Animals used in this study were produced in the AAALAC International accredited Stowers Reptile and Aquatics Facility in compliance with protocols approved by the Institutional Animal Care and Use Committee. They descended from breeding stock collected in New Mexico under permit numbers 3199 and 3395 and Arizona under license number SP564133. Tissues used in the RAD-seq analyses were derived from samples collected under permits from the Arizona Game and Fish Department, California Department of Fish and Wildlife, New Mexico Department of Game and Fish, and the Secretariá de Medio Ambiente y Recursos Naturales, Dirección General de Fauna Silvestre of Mexico.

## Decision letter and Author response

Decision letter https://doi.org/10.7554/eLife.97035.sa1
Author response https://doi.org/10.7554/eLife.97035.sa2

# Additional files

## Supplementary files

- Supplementary file 1. *A. marmoratus* genome assembly statistics.
- Supplementary file 2. Trinity assembly statistics.
- Supplementary file 3. MAKER2 summary.
- Supplementary file 4. Information for all *A. marmoratus* animals sequenced.
- Supplementary file 5. Whole genome sequencing alignment statistics.
- Supplementary file 6. Animals confirmed by microsatellite analysis to be of facultative parthenogenesis (FP) origin.
- Supplementary file 7. Microsatellite primer information.
- MDAR checklist

## Data availability

All raw sequencing data pertaining to the AspMar1.0 genome assembly are available at the National Center for Biotechnology Information under project accession number PRJNA360150. All other whole-genome and RNA sequencing data can be found under PRJNA980964. RAD-seq data can be located under PRJNA827355, PRJNA707030, PRJNA762930, and PRJNA1016487. Code and raw microsatellite data used for analysis are available at GitHub (copy archived at *baumannlab, 2024*).

The following datasets were generated:

| Author(s) | Year | Dataset title | Dataset URL | Database and Identifier |
|---|---|---|---|---|
| Baumann P | 2020 | Aspidoscelis marmoratus isolate:SIMRID8450 (Marbled whiptail) | https://www.ncbi.nlm.nih.gov/bioproject/PRJNA360150 | NCBI BioProject, PRJNA360150 |
| Ho DV, Tormey D, Odell A, Newton AA, Schnittker RR, Baumann DP, Neaves WB, Schroeder MR, Sigauke RF, Barley AJ, Baumann P | 2024 | Sequencing of Aspidoscelis maramoratus | https://www.ncbi.nlm.nih.gov/bioproject/PRJNA980964 | NCBI BioProject, PRJNA980964 |
| Ho DV, Tormey D, Odell A, Newton AA, Schnittker RR, Baumann DP, Neaves WB, Schroeder MR, Sigauke RF, Barley AJ, Baumann P | 2024 | Post-meiotic mechanism of facultative parthenogenesis in gonochoristic whiptail lizard species | https://www.ncbi.nlm.nih.gov/bioproject/PRJNA1016487 | NCBI BioProject, PRJNA1016487 |

The following previously published datasets were used:

| Author(s) | Year | Dataset title | Dataset URL | Database and Identifier |
|---|---|---|---|---|
| Barley AJ | 2021 | Evolutionary study of whiptail lizards (Aspidoscelis) | https://www.ncbi.nlm.nih.gov/bioproject/PRJNA707030 | NCBI BioProject, PRJNA707030 |
| Barley AJ | 2021 | Genetic diversity and the origins of parthenogenesis in the teiid lizard Aspidoscelis laredoensis | https://www.ncbi.nlm.nih.gov/bioproject/PRJNA762930 | NCBI BioProject, PRJNA762930 |
| Barley AJ, de Oca ANM, Normal L, Robert CT | 2022 | The evolutionary network of whiptail lizards reveals predictable outcomes of hybridization | https://www.ncbi.nlm.nih.gov/bioproject/PRJNA827355 | NCBI BioProject, PRJNA827355 |

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
