## [Editor Report]

In this valuable paper, convincing evidence is provided for the production of facultatively parthenogenetic whiptail lizards through a gametic duplication. The audience for the work will be broad, given that parthenogenesis is such a fascinating topic.

---

## [Decision Letter]

[Editors' note: this paper was reviewed by Review Commons.]

Thank you for resubmitting your work entitled "Post-meiotic mechanism of facultative parthenogenesis in gonochoristic whiptail lizard species" for further consideration by *eLife*. Your revised article has been evaluated by Detlef Weigel as Senior and Reviewing Editor.

*Reviewer #2 (Recommendations for the authors):*

The authors have responded appropriately to my suggestions. There is just one lingering area of concern in lines 209-219. First, the authors refer to defining heterozygous sites as having two alleles supported by the "same" number of reads (changed from "equal"), but what does "same" mean? Presumably it's not the exact same number of reads, because this would exclude all sites with an odd number of reads. So how is "same" defined? An analogous issue applies to the "coverage equal to the mean sequencing depth". What "equal" needs in this context needs to be precisely defined.

---

## [Author Response]

1. General Statements

We thank the three reviewers for their thoughtful and constructive comments. The changes to the text and figures made in response to the questions raised have made this a clearer and stronger manuscript. The additional citations suggested by the reviewers helped to further anchor our study within the growing literature on facultative parthenogenesis. Below we have responded to each comment in blue. We have added new data to the manuscript (Figure 4C, Figure S10B and Figure S10D).

Point-by-point description of the revisionsReviewer #1 (Evidence, reproducibility and clarity (Required)):1. Summary: Here Ho et al. provide strong molecular evidence for the production of facultatively parthenogenetic whiptail lizards, through a gametic duplication. As evidenced through multiple routes, including microsatellites, WGS, RADseq, and RBC ploidy, and lines of evidence from multiple specimens, this study is timely in furthering our understanding of the mechanisms underlying FP. The findings are conclusive.That said, I have several comments that should be addressed prior to publication. The introduction which addresses FP in other systems fails to cite several key studies that provide strongly molecular support for terminal fusion automixis. Similarly, the study pushes the idea that this is an adaptive trait, however without proving that the parthenogens can themselves reproduce, this is a moot point at this stage.That said, my comments are minor. I found this to be an excellent study, well written, comprehensive in methodology, and one that I strongly advocate for publication.

We thank reviewer 1 for referring to our manuscript as an excellent study and strongly advocating for its publication. We concur with his/her points that evidence for automixis in other systems was not sufficiently referenced and that the adaptive trait hypothesis for FP is somewhat speculative. The text has been modified accordingly (see below).

Major comments – None.Minor Comments: Should be addressed.Line 36 – However, data that supports terminal fusion are no longer restricted to microsat data. Studies utilizing RADseq and whole-genome sequencing in snakes and crocodiles have now provided further evidence supporting terminal fusion.See:Booth et al. 2023. Discovery of facultative parthenogenesis in a new world crocodile. Biology Letters. 19, 20230129.Card et al. 2021. Genome-wide data implicate terminal fusion automixis in king cobra facultative parthenogenesis. Scientific Reports. 11, 1-9Allen et al. 2018. Molecular evidence for the first records of facultative parthenogenesis in elapid snakes. R. Soc. Open. Sci. 5, 171901.

We have now included that automixis in other systems is supported by both microsatellite and NGS data in the abstract of our manuscript. The references have been included in the main text.

Ln 42 – Evidence suggesting that isolation from males was not a pre-requisite for FP has previously been reported in snakes.See:Booth et al. 2011. Evidence for viable, non-clonal but fatherless Boa constrictors. Biology Letters. 7, 253-256.Booth et al. Facultative parthenogenesis discovered in wild vertebrates. Biology Letters. 8, 983-985.Booth et al. 2014. New insights on facultative parthenogenesis in pythons. Biol J Linn Soc. 112, 461-468.

Despite the prior evidence to the contrary cited by the reviewer, it is still a commonly held belief among scientists and science journalists that isolation from males promotes or triggers FP. We have placed our findings in the context of other studies, including those mentioned above, that came to the same conclusion that isolation from mating partners is not a requirement for FP. We thank the reviewer for the additional citations, which are now included in the Discussion section.

Ln 48 – Is this really an argument. While an immediate transition to homozygosity will purge some deleterious alleles, given the genome-wide nature of this, there will also conversely have been strong selection for mildly deleterious alleles.

Even though many FP animals have congenital defects, our data, combined with that of others, show that seemingly healthy animals arise as well. Even if these healthy animals harbor slightly deleterious alleles, the most detrimental alleles would have therefore been purged especially for subsequent generations. We have modified the abstract to be clearer: “Conversely, for animals that develop normally, FP exerts strong purifying selection as all lethal recessive alleles are purged in one generation.”

Ln 56 – I would recommend the inclusion of both Allen et al. 2018. R. Soc. Open Sci, and Card et al. 2021. Sci Reports, here, as they are members of the elapids, not represented in the other examples.

These two citations have been added.

Ln 60 – Recent studies have highlighted the significance of sperm storage in reptiles. For example, Levine et al. 2021. Exceptional long-term sperm storage by a female vertebrate. PLos ONE. 16(6).e0252049, describe the storage of sperm by a female rattlesnake for ~70 months, with two instances of its utilization to produce healthy offspring during that period. Clearly, molecular tools are providing both support for long-term sperm storage, and an understanding of its utilization.

Recent work has indeed provided new evidence for instances of long-term sperm storage and the two mechanisms are no longer competing hypotheses, but it is clear that both mechanisms exist in nature. We have modified the text accordingly to include “Nevertheless, clear examples of long-term sperm storage have also been documented in the recent literature (29), underscoring the need for molecular methods such as MS analysis or sequencing data to elucidate the underlying mechanisms.”

Ln 68 – American Crocodile would also be suitable to include here.

This has now been included in the list of examples of endangered species.

Ln71 – The problem with this hypothesis is that parthenogens produced through FP tend to have very low viability. For example, Adams et al. 2023. Endangered Species Research, follow a cohort of sharks produced through FP and all survive. Similarly low levels of survival are reported across other systems for which FP was reported. More likely, FP is simply a neutral trait. The mother is not negatively impacted through producing parthenogens and can go on to produce sexual offspring. Few instances report successful reproduction of a parthenogen. See pers. Comm in Card et al. 2021. And Straube et al. 2016.

We thank the reviewer for the comment and agree that more data on the successful reproduction of parthenotes are needed to claim that FP is an adaptive trait. We have modified the text to include that studies on “the successful reproduction by FP offspring” are needed to support this hypothesis and have included the Straube et al. 2016 citation. We decided to omit the Card et al. 2021 citation as the reports of second-generation FP was through personal communication mentioned in this study and the results themselves have not yet been published.

Ln 79 – I doubt that there is a desperate need for this for conservation. However, I think there is a need to simply further our understanding of basic biological function, given that it is not uncommon, and is phylogenetically widespread in species lacking genomic imprinting.

We agree that understanding FP as a basic biological function is important in light of the realization that it occurs more commonly than previously thought. We have added this aspect to the text: “A better understanding of the triggers and molecular mechanisms underlying FP and the fitness of the resulting offspring are therefore needed in a variety of contexts. These include: to understand a fundamental biological mechanism and its significance in vertebrate evolution, to aid in conservation efforts including captive breeding programs, and to possibly harness FP in an agricultural context (28).”

Ln 85 – It would be worth citing Card et al. 2021., here given that they used genome-wide ddRAD markers to show support for terminal fusion.

The citation has been added.

Ln 91 – Better citations here are Card et al. 2021. Allen et al. 2018, and Booth et al. 2023, which all utilize either RADseq or WGS.

These citations have been added.

Ln 95 – The conclusion of genome duplication here was supported only by a small number of microsatellite loci. As such, given that terminal fusion has been supported through genome-wide markers in other species of snakes and crocodiles, the conclusion of genome duplication is likely incorrect.

In light of the other examples that show terminal fusion in snakes, we have removed this sentence.

Ln 96 – I would strongly disagree with this statement. Allen et al. 2018, Card et al. 2021, Booth et al. 2023, all provide evidence of heterozygous loci and thus support terminal fusion. While no species-specific chromosome level reference genome is available for any of these species, the fact that levels of heterozygosity are below 33% percent supports terminal fusion. Rates over 33% support central fusion, but have not been reported in any vertebrate to date. AS such, I would recommend the removal of this statement.

We agree that the studies listed by the reviewer all support terminal fusion in snakes and crocodiles and therefore, we have removed the statement.

Ln 121 – Recent work in Drosophila mercatorum and D. melanogaster suggest that three genes play a role in the activation of FP in unfertilized eggs. In this case, through the fusion of meiotic products. That said, it is plausible to assume that FP in these lizards has an underlying genomic mechanism that is not related to isolation from males. See Sperling et al. 2023. Current Biology. 33, P3545-P3560.E13.

Clearly isolation from males is not a key trigger in FP in whiptail lizards and other vertebrate species. With recent work from Sperling et al. 2023 and the fact that selection has led to increases in parthenogenesis in birds, an underlying genetic mechanism may well be at play. We have cited and addressed this in the discussion and propose identifying the genetic basis for FP in whiptail lizards in future studies.

“Recent work identifying key cell cycle genes inducing FP in two species of *Drosophila* (71) and selection resulting in higher incidences of parthenogenesis in birds (24, 33) suggest a genetic basis for the initiation of FP. […] Additional whole-genome sequencing data for species with documented FP will aid in the understanding the genetic basis, propensity, and evolutionary significance of FP.”

Ln 126 – While these data strongly support FP of the two unusual A. marmoratus appearing offspring, can long term sperm storage be ruled out. Either through captive history or allelic exclusion of other males in the group?

We have added the following sentence to the text: “Given that all of these offspring are female, inherited only maternal alleles, and animal 122 had no history of being housed with a conspecific male during its lifetime, both interspecific hybridization and long-term sperm storage are all but ruled out and FP is strongly supported.”

Ln 171 – 191 – Given that the topic of this manuscript is the genomic mechanism underlying FP in this species, are these data necessary? These are not discussed later and as such I would recommend that they are moved supplemental material. Otherwise, they simply clutter that manuscript and detract from the key question. Indeed, they are important to show that the genome constructed is of high quality, but online Supp Mat is the place for that here.

We chose to keep this section in the main text for the following reasons: There is still a lack of published reference quality genomes for many reptile species and therefore we want to highlight that this *A. marmoratus* reference adds not only to the understanding of FP, but also expands the small list of reptile genomes and makes the first *Aspidoscelis* genome available to the community. The high quality and contiguity of the genome (as indicated by the high N50 value and BUSCO score) is important to emphasize in the main text because the absence of any heterozygous regions in FP animals supports a mechanism of post-meiotic genome duplication. We would not want to bury these key points in the supplement.

Ln 296 – Comparable estimates were made for parthenogenetic production in wild populations of two North American pitviper species. See Booth et al. 2012. Biology Letters.

In Booth et al. 2012, 2 out of 59 litters of the two pitvipers (3.39%) were identified to contain FP offspring and these results are very similar to our reported rate of FP in whiptail lizards. We have now included this similarity in our discussion. “Interestingly, these rates are similar to what has been reported for wild populations of two North American pitviper species (10)”.

Ln 312 – Again, can this really be suggested? Above, the authors state that most FP animals that hatched had congenital defects, and a large number failed to hatch. This does not sound like strong support for generating individuals that counter the effects of population bottlenecks and inbreeding depression. The authors need to take this study further and monitor the long-term viability of the FP individuals that survive.

We agree with the reviewer that the adaptive advantages of FP reproduction are dependent on the fitness and reproductive potential of FP offspring and present data is insufficient to clearly support this notion. We have modified the text to include that long-term studies are needed to support or refute this hypothesis: “However, support for this hypothesis is predicated on the fitness and reproduction of FP offspring and therefore more long-term studies on seemingly healthy individuals of FP origin are needed.”

Ln 348 – To be able to provide support for this, you need to track animals long term to understand their reproductive competence, and that of their offspring.

We have added the text: “To assess whether the co-occurrence of sexual and FP reproduction in vertebrates can indeed be considered a reproductive strategy rather than biological noise will require further studies to assess the reproductive competence and fecundity of offspring produced by either mode of reproduction.”

Ln 358 – But, the caveat is that the parthenogens must themselves reproduce. This must me stated.

The statement that parthenogens must be able to reproduce to support a hypothesis of FP as an adaptive trait has been added: “One must now consider the possibility that FP is an adaptive trait and that low rates of successful FP could contribute significantly to genome purification. Such a role for FP hinges on further studies demonstrating the ability of parthenogens to reproduce themselves either through further FP or sexually.”

Ln 359 – Note that FP can also fix mildly deleterious alleles. Only if it is strongly deleterious will it be lost.

We now make it clearer that selection only applies to strongly deleterious alleles.

Ln 361 – See above comments.

We have modified the text to include that “FP offspring will have low genetic load and only pass on neutral and mildly-deleterious alleles to the next generation.”

Reviewer #1 (Significance (Required)):2. Significance:While reports of parthenogenesis have been reported as far back as the early 1900's, it has only been over the last decade that reports are become common. Such that facultative parthenogenesis is no longer considered a rarity, but is recognized now as being relatively common and phylogenetically widespread in species that lack genomic imprinting – particularly reptiles, birds, and sharks. Reasons for this are both an increased understanding that the trait can occur, hence recognizing it as an alternative mechanism to long-term sperm storage, and the ease of using molecular approaches.The fundamental questions of recent times have been understanding the mechanisms driving FP. Recent papers utilizing whole genome sequencing and ddRADseq have provided support for terminal fusion automixis in snakes and sharks. Here, this study provides evidence of gametic duplication in whiptails, a mechanism with an alternative outcome in regards to the levels of retained heterozygosity. As such, this study compares to the recent work of Card et al. 2021 (Scientific Reports), and Booth et al. 2023 (Biology Letters), in providing substantive advances in the field.The audience for this will be broad. Parthenogenesis is a fascinating topic that attracts significant media attention. See the Altmetric score of recent papers on the topic, particularly Booth et al. 2023 (Altmetric score – ~3100). As such, the study will be of interest to both a broad readership, but will also be of great significance to a specialized group working on parthenogenesis. All round, an excellent paper that has promise to advance the field.

We thank reviewer 1 for this positive assessment and for putting our work into context.

Reviewer #2 (Evidence, reproducibility and clarity (Required)):Summary: The researchers bring together microsatellite and whole-genome sequencing data from long-term laboratory cultures of lizards to discover occasional production of parthenogenetic offspring by several species of otherwise sexually producing whiptail lizards ("facultative parthenogenesis, "FP") and to show that these FP-produced lizards have patterns of genomic homozygosity that are incompatible with currently held assumptions about mechanisms of FP. Instead, the FP lizards seem to have been produced by a mechanism that results in almost complete homozygosity, likely a consequence of post-meiotic duplication of genomes from haploid unfertilized oocytes. They also show that FP offspring were produced by females housed with males and along with sexually produced offspring, counter to prevailing assumptions that FP offspring are only produced in situations where mates are not available. Many of the FP-produced offspring did not survive to hatching or had major abnormalities, consistent with a situation where this high homozygosity exposes harmful alleles. Finally, the authors used reduced-representation sequencing (RAD-seq) to survey heterozygosity in 321 wild-collected whiptail lizards from 15 species, showing evidence for strikingly low homozygosity in at least one individual and perhaps up to 5, consistent with the potential for FP in nature. These data are of broad interest in demonstrating several exciting new possibilities. Most importantly, the data hint at a different mechanism of FP than previously assumed, and one that causes immediate near-complete homozygosity. This scenario would likely lead to immediate purging of harmful recessive alleles. If the selective load of this purging wasn't insurmountably high, a lineage with a history of purging could produce FP offspring of relatively high fitness. Other exciting possibilities suggested by the data include the existence of FP even in a setting where mating occurs and in natural populations, versus just captivity.Major Comments:I found it difficult to impossible to sort out exactly what the researchers did and with what lizards. For example, in line 107, they refer to a "systematic MS analysis" for all individuals of gonochoristic species in their laboratory, but where are these data? Indeed, at this early spot in the paper, the introduction from here on out suddenly reads like a discussion. What would be better here would be to summarize what was known and wasn't known about the system and questions involved, why gaps in knowledge were important, and what the researchers actually did for this paper. In my opinion, the paper would be a much easier read if the researchers left the results and interpretation for later in the paper.

As a consequence of the reviewers’ comments, the text of the manuscript has undergone major revision, and we trust that reviewer 2 will find this new version far more accessible. The MS data collection of more than 1000 individuals is the subject of another ongoing study and was only mentioned peripherally here to put the identification of FP into context. As most of the MS data relates to gonochoristic reproduction and interspecific hybridization, we are only presenting the data that are directly relevant to this manuscript as part of this study. To our knowledge, there is no common repository to upload raw MS data, but we have provided the data for the FP animals and controls discussed in this paper in the Github repository (see section “Data availability”).

Even with this suggested fix, however, the data are still too inaccessible and analyses too opaque. For example, in line 202, a critical definition is laid out regarding heterozygous sites as those having "equal support" for two alleles. What do the researchers mean by "equal support"? My presumption is that this is something about equal or close to equal numbers of reads, but this definition needs to be spelled out and justified because it underpins much of the downstream analyses. A similar problem occurs in line 208-209, where the authors make a statement about limiting further analysis to positions in the genome where the coverage is "equal" to the mean sequencing depth.

We have changed the text to “we defined heterozygous sites as those having two alleles supported by an equal number of reads. This stringent requirement was chosen to limit the search to apparent heterozygous sites with strong support, decreasing the chance of false positives.”. We further look at only sites where the coverage is equal to the average sequencing depth to exclude regions where over-assembly and collapse of repetitive elements would artificially increase the coverage.

Another data/analysis issue emerges with the components of the manuscript that deal with mixoploidy. As far as I can tell, these data come from one sexually produced lizard, one FP A. marmoratus, and one FP A. arizonae. While the reports of bimodality of nuclear size are certainly interesting, the data and discussion are no more than an anecdotal case study in the absence of careful replication across multiple FP lizards and comparison to sexually produced lizards. Without these data, the conclusion that “Animals produced by facultative parthenogenesis are characterized by mixoploidy” (Figure 4 caption; also see lines 324-331) is far too strong.

We have added animal IDs to figure legends 4 and S10 to clarify that these erythrocyte staining come from two FP *A. marmoratus*, and one FP *A. arizonae*. In addition, imaging from two sexually produced control animals (1 *A. marmoratus and* 1 *A. arizonae*) have now been included in S10 (as S10B and S10D). We also have included an extra panel of flow cytometry data (new Figure 4C) as a complementary methodology for ploidy determination. Both imaging and flow cytometry support similar amounts of haploid cells. With the additional data and clarification, we hope that the reviewer agrees that the observations of mixoploidy are well beyond “anecdotal”. Nevertheless, we have changed the title for Figure 4 to “Detection of mixoploidy associated with facultative parthenogenesis.” We hope that our observations here will indeed inspire future studies to see if mixoploidy is a widespread phenomenon in FP outside of whiptails as indicated by earlier work in birds.

I had a similar reaction to the discussion of developmental abnormalities and embryonic lethality of embryos of FP origin presented in lines 263-281 (also lines 307-309). What is the baseline level of such abnormalities and the frequency of lethality in sexually produced eggs/embryos/hatchlings, and especially those produced via inbreeding? These comparisons are needed to interpret the significance of the patterns observed in the FP eggs/embryos/hatchings. Analogously, the comparison of the ovaries and germinal vesicles from one FP individual relative to one sexual individual do not tell us anything nearly so definitive as the text in lines 279-281 (also see Figure S12 title, which is too broad of a conclusion for N = 1). This overly ambitious conclusion also underpins the discussion regarding the potentially adaptive nature of FP with respect to genome purification (lines 341-363; also see lines 47-50). If FP does not actually increase the rate of purging in FP lizards relative to inbred sexual counterparts (sounds like inbreeding is common from line 339), it seems less likely that we can view FP as adaptive at least from this perspective.

We have now included a comparison between defects seen in sexually produced animals vs FP animals: “six out of 16 FP animals (37.5%) hatched with no discernable developmental defects (Figure S11A-B). This is in stark contrast to sexually produced animals, where over 98% of hatchlings showed no abnormalities. Additionally, most of the defects noted in sexually produced animals were less severe than in FP animals including bulges in tails or truncated digits.”

We agree that our statement on the lack of differences between sexually produced and FP animals was too general. We have modified the title of Figure S12 from “No differences between ovaries and germinal vesicles of *Aspidoscelis marmoratus* produced by facultative parthenogenesis or fertilization” to "Ovaries of Aspidoscelis marmoratus FP animal 8450 and germinal vesicles of FP sister 8449 revealed no differences in structure and anatomy compared to fertile sexually reproducing animals.” Due to instant complete homozygosity, FP would indeed have a higher rate of purging than inbreeding. While one hypothesis is that FP is adaptive (in large enough populations), our intentions were to highlight the alternative that FP could be detrimental in smaller populations (that already would likely experience high inbreeding rates). We would expect inbreeding to not be common in whiptails relative to other lizards given that they tend to have large population sizes and actively range across generalist habitats.

A final data concern is with the use of liver tissue for whole-genome sequencing and reference genome assembly (lines 389-390) and then using these data and the reference genome to make conclusions about ploidy/coverage. Liver tissue is very commonly endopolyploid, meaning that coverage could be artificially high for animals for which liver (vs. tail) tissue was used for DNA extraction. In particular, it would be helpful if the researchers consider whether endopolyploidy could have affected their ability to make accurate estimation of coverage and thus, heterozygosity, when libraries generated from diploid (tail) tissues are aligned to a reference genome generated from a polyploid tissue as was done here.

This is an interesting point and indeed hepatic cells in various organisms have been documented to be polyploid. The proportion of polyploid cells though vary and as far as we are aware, all published studies on polyploid hepatocytes are in mammals (DOI: 10.1016/j.tcb.2013.06.002). Reference genomes have been generated from a variety of tissue sources and liver is commonly used. As most assemblies are for haploid genomes, polyploidy (unlike aneuploidy) does not impact the assembly quality. The reference genome was also from an animal of FP origin and therefore has genome-wide homozygosity that aids in a more contiguous genome assembly by eliminating the phasing problem. For the 10 animals sequenced, genomic DNA was derived from liver for three animals and the rest from tail tissue. The sequencing data generated from either liver or tail resulted in similar coverage levels (Figure S6) and similar levels of heterozygosity (Figure 2A).

Minor Comments:Line 410: Please explain why the BLAST cutoff was changed from the default.

The BLAST cutoff was changed from the default 1e-03 to 1e-06 to be more stringent and thereby increase confidence in the BUSCO results.

Lines 441-443: Please explain why this dataset was seemingly larger than expected.

Animal 122 was sequenced on one flow cell without any multiplexing with other samples and therefore yielded more reads than other animals sequenced. We subsampled the reads from this animal for analysis, so it is directly comparable with the other WGS data.

Line 510: The link to the Github repository was broken, so I was unable to access the code and data denoted as available here.

We apologize for the unavailability of the link at the time of review. Review Commons did not request a reviewer token. The repository will be made public upon journal acceptance. We would be happy to provide a reviewer token in the meantime upon request by Review Commons.

Figure 1, and other figures featuring comparisons of MS data across parents and offspring: The authors need to engage here with the alleles that do not match either parent here (e.g., allele 282 at MS7), explaining the likelihood that these alleles indeed represent a binning error (or, perhaps, stepwise mutation from parental allele), and these alleles should be flagged. Instead, they bin these unique alleles with the most similar parental allele without any explanation or flagged. The authors do bring this point up in Figure S1, but this issue needs to be addressed in the main text (related point: the mix of red/green in MS16 offspring appear more green than red. Is this meant to denote a probability different than 50:50? If not, the authors should adjust the shading so that this shape is half green, half red).

We have added to the figure legend that single nucleotide differences are most likely binning errors and are therefore not considered “de novo” alleles. Instead, they are assigned it to the most similar parental allele, consistent with Figure S1. The shading at MS16 has been removed so that it is consistent with Figure 3.

Figure 3: Indicate that white background for alleles means that allelic inheritance is not determinable, or use the mix of colors applied in Figure 1 to indicate as such. Unique offspring alleles should be flagged rather than just automatically assigned to the most similar parental allele. Finally, it would be helpful if the alleles were presented within loci from the shorter to the longer alleles.

We have included in the figure legend that non-shaded alleles are those for which multiple potential parents share the same allele and the inheritance therefore remains ambiguous for this locus. Single nucleotide differences are also now addressed, and sizes are ordered from smallest to largest.

Figure S7. Indicate visually which panels indicate FP animals.

We have now indicated which animals are FP and included this in Figure S6 as well.

Figure S13. The 5 animals that had especially low heterozygosity should be flagged. The title of this figure should be toned down in light of the tentative nature of the conclusions regarding FP in nature: low heterozygosity could instead reflect, for example, a long history of inbreeding. My reaction to the data is also that the % heterozygosity distribution for many of the species looks continuous rather than the bimodality one might expect under FP vs. sexual reproduction.

Since FP has not been further confirmed in these animals, unlike those examples from our captive colony, there could indeed be other reasons for low heterozygosity. We have changed the title of the figure from “Facultative parthenogenesis in whiptail lizards collected in nature” to the more neutral “Heterozygosity estimates of whiptail lizards collected in nature.” Since there are so relatively few animals, one would not necessarily expect a bimodal distribution to be apparent in the current data. We did show that the animal with the lowest calculated level of heterozygosity (*deppii* LDOR30) was a statistical outlier when compared to other individuals of the same species though. Since these animals were sampled across different locations and habitats, the effective population sizes would be assumed to be different as well, reflecting the range of heterozygosity estimates seen here. This has been made clear in the text.

Reviewer #2 (Significance (Required)):General assessment: strengths and limitations. The paper's strengths include the combination of data from lab and natural populations, the characterization of an unexpected means of achieving FP, with dramatic genetic consequences, and the data suggesting that this type of FP is fairly common and occurs even in the context of mating.Audience: The biological questions of relevance to these discoveries are of broad interest, and the paper is likely to garner some attention from the life sciences community as whole and the popular press.Advance: These data fill an important knowledge gap regarding the mechanisms potentially driving FP in vertebrates, how often FP is likely to occur, and its genetic consequences. The discoveries are potentially conceptual/fundamental, though the extent to which they are ground breaking is not clear in the absence of functional characterization of how FP occurs as well as the need for more rigorous comparisons and replication that I outlined above.

We thank reviewer 2 for summarizing the strengths of this manuscript, pointing out the broad interest and stating that this work fills an important knowledge gap.

Reviewer #3 (Evidence, reproducibility and clarity (Required)):Summary:The occurrence of facultative parthenogenesis has been described in a number of vertebrate lineages but the underlying cytological mechanism(s) have remained largely speculative due to sparsity of data. Here, Ho & Tormey et al. provide a detailed analysis of facultative parthenogenesis in gonochoristic species of the lizard genus Aspidoscelis. They show that parthenogenesis leads to a complete loss of heterozygosity (LOH) within a single generation. They attribute the LOH to diploidization through duplication of the oocytes haploid genome after completion of meiosis. This mechanism is consistent with their finding of mixoploidy in erythrocytes of asexually produced offspring. Based on LOH the authors additionally show that facultative parthenogenesis in Aspidoscelis is not condition dependent (no developmental switch): it can occur in the presence of males, alongside with sexual reproduction in the same clutch, and both in captivity and the wild. Finally, the authors show that facultative parthenogenesis is associated with developmental aberrations, likely caused by expression of homozygous recessive deleterious mutations.Major comments:In my opinion, this study presents a very comprehensive, careful documentation of mechanistic aspects and consequences of facultative parthenogenesis in a vertebrate. The genomic and microsatellite results leave little to no doubt that facultative parthenogenesis has led to complete LOH in Aspidoscelis. I am particularly impressed by the meticulous analysis of genomic coverage to exclude e.g. false positive heterozygosity due to merged paralogs in the assembly. I also follow the authors conclusion that a post-meiotic "gamete duplication"-like mechanism is likely causative for the LOH (and the mixoploidy of erythrocytes; but I am no expert on that). I was wondering if terminal fusion automixis together with a complete absence of recombination would be worth mentioning as an (probably very unlikely) alternative in the discussion. It would be exciting to corroborate the conclusion of diploidization by genome duplication in the future, e.g. via early embryonic DNA stainings to show the duplication "in action" (if that is practically possible)…? As for this manuscript, I suggest emphasizing the indirect nature of the evidence for the mechanism of parthenogenesis a little bit more.

We thank the reviewer for highlighting the effort that went into the genomic analysis that led us to our conclusions. In terms of terminal fusion without recombination, we argue that this is not an obvious alternative explanation as a large body of work has established that at least one crossover per homologous chromosome pair is required to advance into meiosis I in many organisms (e.g. see https://doi.org/10.3389/fcell.2021.681123) and therefore the absence of recombination would likely not produce the polar bodies necessary for automixis.

We have added to the text: “In whiptail lizards, we have not been able to examine post-meiotic oocytes as locating the post-meiotic nucleus within a large yolked egg is inherently difficult. The difficulty is compounded by the unpredictability of which eggs will undergo FP development and the need to sacrifice animals to remove eggs.”

While the genome duplication mechanism we propose is indeed indirect because we are unable to visualize developing FP embryos, the most parsimonious explanation from the whole-genome sequencing analysis is genome duplication because of the lack of heterozygous regions associated with automixis. In the text, we have made sure to state genome-wide homozygosity as the basis for our conclusion.

I agree that facultative parthenogenesis in the presence of males hints at a baseline rate of parthenogenesis without requiring a developmental switch. However, this makes it difficult to rule out that sperm played a role in activation of embryonal development (gynogenesis; however I am only aware of gynogenesis in fishes and amphibians)… maybe, the authors want to take this up in the discussion. Were the five parthenogenetic individuals for whole genome sequencing actually produced in the presence of males, too?

FP has been reported to occur in isolated females for other reptile and bird species, suggesting that sperm activation is at least not a general requirement in FP of amniotes. (Watts, et al. 2006, W. W. Olsen, S. J. Marsden 1954). In all cases in this study, the female mothers were housed with conspecific or heterospecific males. While we cannot completely rule out a non-genetic contribution of sperm in these cases, it would seem to be an unlikely explanation in light of the sperm-independent reproduction by obligate parthenogenesis in other species of whiptail lizards (unlike the sperm-dependence of all unisexual reproduction in amphibians and fish). We decided to not include speculation on sperm-dependence in this manuscript as we have no evidence in favor of it, nor is there any evidence for this in the literature relating to other amniotes. In fact, most examples of FP were reported from isolated females, most likely because offspring were not expected in those cases and prompted further analysis as to their origin.

I agree with the interpretation of the LOH in the RADseq data as a likely case of facultative parthenogenesis in the wild. However, when looking at figure S13 I noticed some bimodal looking distributions (e.g. in A. guttatus). It may be interesting for future studies to look into what factors influence heterozygosity in natural populations of Aspidoscelis (e.g. inbreeding vs parthenogenesis). Could there be different mechanisms of facultative parthenogenesis in different Aspidoscelis species explaining different LOH intensities?

The continuous nature of the data may reflect natural variation between individuals and collection at various locations with possibly different effective population sizes and levels of hybridization. Low levels of heterozygosity could be indicative of inbreeding or FP in some cases. This is important to note in future studies and we have added this to the manuscript (“Further fieldwork and analysis will be required to assess the level of FP in natural populations of gonochoristic *Aspidoscelis* species (and other factors that could influence the observed heterozygosity such as population size, levels of hybridization, and inbreeding) …”). While there are different mechanisms of FP in other vertebrate groups, the most parsimonious hypothesis is that within a genus, the mechanism would be the same.

The manuscript is well written, the introduction nicely explains the significance of the study, the methods are fully appropriate and the results (and supplementary results) displayed comprehensibly and in great detail. The discussion might benefit from going a bit more generally into the occurrence and mechanism of obligate asexuality in Aspidoscelis. One might e.g. speculate on whether the ability for facultative parthenogenesis in gonochoristic species has facilitated the transitions to obligate parthenogenesis in the hybrid lineages and what peculiarities might predispose Aspidoscelis to parthenogenesis (e.g. are centrioles contributed by sperm required?). In addition, I think the occurrence of LOH due to gamete duplication (facultative and obligate) in invertebrates (e.g. due to Wolbachia) is worth mentioning in the discussion: e.g. there is a similar case in facultative asexual Bacillus rossius stick insects, where the early dividing cells are haploid. Some of them diploidize via duplication later and form the embryo.

Thank you for complimenting each section of the manuscript and referring to it as well-written. Our lab has a long-standing interest in obligate parthenogenesis. While it is interesting that both obligate and facultative parthenogenesis occur alongside each other in this genus, the mechanisms appear to be fundamentally different, and we would like to focus the discussion on FP in a variety of systems and its potential implications in conservation and evolution. Parthenogenesis in general is a fascinating topic for a broad audience and not discussing another form of parthenogenesis (obligate in this case), the focus remains on FP and keeps the manuscript more accessible for non-specialists. We have included the stick insect as another example of diploid restoration through genome duplication in the discussion.

Minor comments:39-41: I am a bit puzzled by the usage of the term "post-meiotic" to contrast the diploidization through duplication with automixis. Wouldn't one consider polar body fusion after completion of meiosis II also post-meiotic? Maybe I am just not aware of how the term is usually used in this context here…

We use the term “post-meiotic” because the restoration of an entirely homozygous diploid cell can only occur after the completion of both meiotic divisions. It is our understanding that polar body fusion and meiotic restitution after meiosis I or meiosis II are generally considered meiotic mechanisms in the specialized literature, even though polar body fusion would also occur after the meiotic divisions.

65: isn't that gynogenesis (sperm-dependent parthenogenesis) in the amazon molly?

While sperm is required for parthenogenesis in the Amazon Molly, it is an all-female species that exclusively reproduces through gynogenesis. In this case, it is considered an example of obligate parthenogenesis rather than FP.

78: the term "economically viable" may be a bit puzzling for a biologist's audience. "Economically sustainable" could be an alternative.

This has been changed.

129: the Arizona male was referred to as ID 4272 above. Here it is ID 4238?

This has been corrected. The correct ID is 4272.

218: please define over-assembly (see line 207)

The definition of “over-assembly” is collapsing paralogous loci into a single representative sequence. This is now explained in the text.

263-281: please, indicate a hatching rate/ rate of malformations of sexually produced offspring for comparison.

A comparison has been added: “This is in stark contrast to sexually produced animals, where over 98% of hatchlings had no abnormalities noted.”

333: in the haploid cells recessive deleterious mutations would be exposed in the hemizygous state but in the diploid cells in the homozygous state.

The text has been modified to reflect the difference between haploid and diploid cells.

470: please, provide more detail for the RADseq analyses (variant calling, calculation of heterozygosity etc.)

We have elaborated on the analysis in the methods.

Figure 1B: please, mention in the legend that the shown mechanisms are not exhaustive, e.g. first polar body fusion could occur right after meiosis 1 or polar body formation could be skipped completely.

This has been added.

Figure 1C: it may be interesting for non-specialists to name the distinctive morphological characters setting apart the three species in the figure legend and highlight them e.g. with arrows in the figure.

We have now included in the figure legend characteristic color patterns for each species: “(C) Photographs of *Aspidoscelis arizonae* with characteristic blue ventral coloration (top), *A. gularis* with light spots in dark fields that separate light stripes on dorsum (middle), and *A. marmoratus* with light and dark reticulated pattern on dorsum (bottom).” Since the descriptions are specific and apparent, we did not add arrows to the pictures.

Reviewer #3 (Significance (Required)):Significance: The study by Ho & Tormey et al. substantially enhances the understanding of (facultative) asexuality in vertebrates. In particular, while most reports of facultative parthenogenesis in vertebrates have been attributed to a form of automixis, the authors conclusively show an instance of diploidization through genome duplication, a mechanism functionally similar to "gamete duplication". The study is novel, very comprehensive and of interest for a general audience within the field of evolutionary biology.

We thank reviewer 3 for pointing out that our study substantially enhances the understanding of asexuality in vertebrates, is very comprehensive and of interest for a general audience within the field of evolutionary biology.

[Editors’ note: what follows is the authors’ response to the second round of review.]

Reviewer #2 (Recommendations for the authors):The authors have responded appropriately to my suggestions. There is just one lingering area of concern in lines 209-219. First, the authors refer to defining heterozygous sites as having two alleles supported by the "same" number of reads (changed from "equal"), but what does "same" mean? Presumably it's not the exact same number of reads, because this would exclude all sites with an odd number of reads. So how is "same" defined? An analogous issue applies to the "coverage equal to the mean sequencing depth". What "equal" needs in this context needs to be precisely defined.

We have addressed the remaining comment from reviewer #2 in the text as follows:

“…we defined heterozygous sites as those covered by an even number of reads with two alleles supported by the same number of reads. Sites covered by an odd number of reads were filtered out for this initial analysis…” (lines 205-107).

Furthermore, we provide an example of “coverage equal to the mean sequencing depth” to further enhance clarity: “Based on this observation, we limited further analysis to positions in the genome where the coverage is equal to the mean sequencing depth (as defined by rounding the mean sequencing coverage value to the next even integer). For example, for animal 003, the average sequencing coverage is 18.31 (Figure 2-supplement 5A) and we only considered sites with a coverage of 20 (Figure 2-supplement 6A)” (lines 213-216). We hope this will sufficiently clarify the approach for reviewer 2 and other readers.